

# Accommodation space indicates dune development potential along an urbanized and frequently nourished coastline

Corjan Nolet[1] and Michel J.P.M. Riksen[1]

[1]Soil Physics and Land Management Group, Wageningen University, P.O. Box 47 6700 AA Wageningen, The Netherlands

**Correspondence:** C. Nolet (corjan.nolet@gmail.com)

**Abstract.** With densely populated areas well below mean sea level, the Netherlands relies heavily on its dunes to ensure coastal safety. About half of the sandy coastline, however, is subject to structural marine erosion and requires frequent sand nourishments as a counteractive measure. A key component of present-day coastal safety policy is creating favorable conditions for natural dune development. These conditions essentially involve a (1) steady supply of wind-blown sand towards (2) wide accommodation space where sand can accumulate and dunes are sheltered from storm surge impact. This paper examines to what extent an experimental mega-scale beach nourishment (termed *Zandmotor* in Dutch) has contributed to creating accommodation space favorable for dune development. Using publicly available airborne Lidar data and Sentinel-2 satellite imagery, favorable accommodation space is identified by comparing recent changes in coastal morphology against dune vegetation cover dynamics. With a focus on European marram grass (*Ammophila arenaria*) as the most prominent dune-building species, this paper demonstrates that the *Zandmotor* supports an especially high potential for incipient (embryo) dunes to develop as most of its favorable accommodation space is located on the beach. However, considering persistent anthropogenic disturbances arising from recreation, leisure and nature management, dune development along this urbanized coastline may not reach its full potential.

## 1 Introduction

Vegetated coastal foredunes, the most seaward facing dune ridge parallel to the coastline, often form the first and foremost line of flood defense against the sea (Hardisty, 1994). This is certainly true for the Netherlands, a country where the most densely populated areas are situated well below mean sea level. And, although widely known for its hydraulic engineering structures, over 75% of the Dutch coastline relies on its foredunes to ensure coastal safety (Van Koningsveld and Mulder, 2004). About half of that sandy coastline, however, is subject to structural marine erosion and requires frequent nourishments of dredged-up sand as a counteractive measure (Van der Wal, 2004). Subsequently, while the coastal foredunes in the Netherlands are considered natural landscape elements, they are so strongly modified by human intervention that they can be regarded a feat of hydraulic engineering in their own right (De Vriend et al., 2015).

The focus of Dutch coastal policy has traditionally been on safety from flooding, which was guaranteed by large structures such as sea dikes, groins and other man-made barriers. Nowadays, however, the focus has widened to also include preserving the spatial quality and natural values of the coastal zone (Van Slobbe et al., 2013). It was recognized that, apart from flood




defense, coastal dunes also represent unique ecological and recreational values and often serve as an important source for drinking water supply (e.g., De Jong et al., 2014; Keijsers et al., 2015b). By acknowledging sand as 'the carrier of all coastal functions', the principal management practice since 1990 has been to nourish the coastline with sand whenever it was about to retreat beyond a defined reference position (Van Koningsveld et al., 2007). A key component of such a dynamic preservation of the coastline involves utilizing natural processes to further redistribute the sand. By allowing marine and aeolian forces to gradually help shape the coastline, the aim of this building-with-nature engineering strategy is to counteract a negative sediment balance while minimizing adverse effects to the coastal ecosystem (Van Slobbe et al., 2013).

An essential element of building-with-nature, is the utilization of European marram grass (*Ammophila arenaria*) to help stabilize and build-up the foredunes (Keijsers et al., 2015b). Compared to other coastal pioneer psammophytes that thrive on sandy soils, this beach grass is most effective at dune-building because not only can it trap high amounts of wind-blown sand in its leaves and roots, but it will in fact grow much more vigorously because of regular burial in sand (e.g., Disraeli, 1984; Van der Putten et al., 1988; Hesp, 1991; Nolet et al., 2017). This introduces a reinforcing feedback crucial to coastal dune development: trapping of wind-blown sand encourages marram grass to grow, which in turn enhances the capacity of marram grass to trap sand and build dunes (Maun, 1998; Zarnetske et al., 2012). Throughout temperate regions in the world, as a result, marram grass helps to create the very high vegetated foredunes critical to coastal safety (Ranwell, 1972). Moreover, because of positive plant-sand feedback dynamics, when the shoreline is frequently nourished with dredged-up sand the adjacent foredunes have the capacity to grow in pace with sea-level rise due to changes in the global climate (Temmerman et al., 2013).

For that reason, present-day coastal safety policy in particular prioritizes creating favorable conditions for natural dune development. These conditions, essentially, involve a (1) steady supply of wind-blown sand towards (2) wide accommodation space (Jervey, 1988) where sand can accumulate and dunes are sheltered from storm surge impact (e.g., Ruggiero et al., 2001; Claudino-Sales et al., 2008; Montreuil et al., 2013; Van Puijenbroek et al., 2017a). Although a predominantly landward wind regime is required, beach morphology arguably exerts the largest control on foredune development conditions (Short and Hesp, 1982). Aeolian sand supply, for example, is typically highest on gently sloping (i.e. dissipative) beaches, where wide intertidal areas store large volumes of sand that are well sorted by wave action (e.g., Hesp, 1989; Anthony, 2013). Wide accommodation space for foredune development is provided when such a dissipative coastline is supported by a wide supratidal beach that is high enough to offer protection from storm surge impact (e.g., Sallenger Jr, 2000; Suanez et al., 2012; Houser et al., 2008). Foredune development is initiated when discrete clumps of incipient dunes (i.e. embryo dunes) form after marram grass establishes itself on the beach via seeds and rhizome dispersal. Over time, and under the right conditions, these individual embryo dunes can then merge into continuous and shore-parallel foredune ridges that extend in a seaward direction (Hardisty, 1994; Hesp, 1989, 2002).

Creating favorable conditions for dune development is an important goal for the *Zandmotor* (Dutch for 'Sand Motor', see fig. 1), an unprecedented mega-scale beach nourishment of 21.5 $Mm^3$ constructed in 2011 just south of The Hague (Stive et al., 2013). The overall purpose of this mega-nourishment experiment is to test whether the benefits (in terms if coastal safety, spatial quality and natural values) outweigh the extra costs involved and to determine to what extent such approach can help cope with expected changes in the global climate (e.g., Mulder and Tonnon, 2011; Stive et al., 2013; De Schipper et al., 2016).



In line with the building-with-nature approach, natural dynamics are encouraged to redistribute the sand of the *Zandmotor* along the coastline, thereby broadening the adjacent foredunes and beach. Specifically, the main objective of the project was defined as: 'Encouraging natural dune growth, primarily in width, in the coastal cell between the cities Rotterdam and The Hague. This creates a larger sand buffer to cope with rising sea level as well as more space for nature and recreation and a larger freshwater lens under the dunes' (Fiselier, 2010; Van Slobbe et al., 2013).

However, while the Delfland coast partly maintains relatively wide and natural dune areas, the aerial photos in fig. 1 clearly show that in some places the dunes are not more than a narrow foredune ridge that is directly bordered by urban areas. Because the region is densely populated, the coastline (including the *Zandmotor*) faces persistent pressure from anthropogenic disturbances. Even though the Delfland coast is meant to serve a wide range of socio-economic functions related to recreation and leisure, activities arising from these function are often in direct conflict with the objectives related to coastal safety and natural values (e.g., Jackson and Nordstrom, 2011; Lithgow et al., 2013). Within this context, this paper examines to what extent the *Zandmotor* has contributed to creating accommodation space favorable for dune development, i.e. accommodation space sheltered from storm impact and experiencing a steady accumulation of wind-blown sand. This favorable accommodation space is identified, using publicly available remote-sensing data, by (1) comparing the presence of dunes against the current morphology of the Delfland coast and (2) comparing recent coastal morphological changes against changes in dune cover by marram grass. Then, by taking the existing dunes into account, the identified favorable accommodation space is used to indicate the full potential for dune development along this urbanized and frequently nourished coastline.

## *FIGURE 1 ABOUT HERE.*

## 2 Materials and methods

### 2.1 Regional setting

The *Zandmotor* is located along the Delfland coast, an approximately 15 km long stretch of coastline that runs between Rotterdam and The Hague parallel to the dominant south-western wind direction. The Delfland coast has a long history of coastal erosion; early 17th century maps make clear that the coastline, compared with today, experienced a significant (> 1.5 km) landward retreat (Van der Meulen et al., 2014). In the late 19th century the Delfland coast was fortified by 64 groynes, but that only slowed down coastal erosion to a landward retreat of about 1 m/y on average. Therefore from the early 1980s onward, well before it became central policy, the Delfland coast has been frequently replenished with sand nourishments of varying volumes (Van Koningsveld et al., 2007). Still, in 2002 the Delfland coast was labeled a 'weak link' as it did not meet stricter coastal safety standards that reflected expected increases in storm surge frequency and magnitude due to climate change (Keijsers et al., 2015b). Between 2009 and 2011, to guarantee for the Delfland coast to withstand hydraulic boundary conditions with a recurrence period of once every 10.000 years, most of the existing foredunes were broadened in a seaward direction (up to about 100 m) with a sand nourishment of approximately 20 Mm$^3$ over a length of 12 km. This included construction of a new dune area in the south (called *Spanjaardsduin* in Dutch) to compensate for loss of natural values because of expansion of



the Rotterdam harbor (Van der Meulen et al., 2014). After completion the newly created dune ridge ranged between 4-12 meter above mean sea level (m +MSL) from toe to crest. For stabilization, the approximately 15° stoss slope was manually planted with marram grass in a regular pattern of about 7-9 small tussocks per m$^2$.

Once the new safety standards were met, the Delfland coast was considered an appropriate location to conduct the *Zandmotor* mega-nourishment experiment. As can be seen in fig. 1, the *Zandmotor* has a hook-shaped design that aims to mirror the natural onshore migration of an intertidal sandbar. Just after construction it had a surface area of about 28 ha, extending 2.5 km along the coastline and protruding 1 km into the sea. Natural processes have since then been working to redistribute the sand, causing the base to flatten and the sand to spread in alongshore directions (De Schipper et al., 2016). Based on an empirical relation between beach width and dune foot migration, a preliminary modeling study estimates that after 20 years the *Zandmotor* would broaden the dunes along the Delfland coast by about 33 ha (Mulder and Tonnon, 2011). A distinct feature that sets the *Zandmotor* apart from more traditionally nourished or natural coastlines, is the locally very high construction height. Most of the base is constructed at a height around 5 m +MSL, while just north of a small dune lake the *Zandmotor* reaches a height of 7 m +MSL. This is well above the current maximum storm surge level of about 3 m +MSL, so reworking of sand on the *Zandmotor* is almost exclusively due to aeolian forcing (Hoonhout and De Vries, 2017). The semidurnal tidal dynamics are characterized by a spring / neap tidal amplitude around 2.0 / 1.5 meter, generating alongshore currents velocities up to 0.5 ms$^{-1}$ (Luijendijk et al., 2017).

## 2.2 Coastal morphology

Data on morphology and morphological changes of the Delfland coast were obtained from Digital Terrain Models (DTM) provided by Rijkswaterstaat, the executive agency of the Ministry of Infrastructure and Water Management. The 2 m pixel size DTM's are produced every year (since 1996) for coastline monitoring purposes by airborne Lidar and have been made public under a Creative Commons Zero (CC0) statement. The contractor responsible for the lidar flights guaranteed a minimum density resolution of 1 point per square meter and a vertical error distribution equal or less than $2.6 \pm 2.0$ cm. Five yearly DTM's of the Delfland coast were used for analysis, acquired in spring 2013 till spring 2017. Changes in coastal morphology were expressed by average change in height per year (m/y) and calculated per consecutive time-step: $t_{5-1} = (t_{2-1} + t_{3-2} + t_{4-3} + t_{5-4})/4$. This was done to better consider temporal variations within each year and to account for yearly changes of the shoreline. Data are available at: https://rijkswaterstaat.nl/apps/geoservices/geodata/dmc.

## 2.3 Dune vegetation cover

Data on dune cover by marram grass were obtained using Sentinel-2A multispectral satellite imagery provided by the European Space Agency (ESA). Sentinel-2 images are acquired in thirteen spectral bands in the visible (VIS), near-infrared (NIR) and shortwave-infrared (SWIR) spectrum. Two level-1C images were downloaded from the Copernicus open access hub (https://scihub.copernicus.eu), with acquisition dates 11-09-2016 and 12-10-2017. With applied radiometric calibration and geometric correction, level 1C images contain top of atmosphere (TOA) reflectance in cartographic geometry (Drusch et al., 2012). Out of the thirteen available bands, the four bands with the highest spatial resolution (10 meter pixel size) were selected





for further analysis. Table 1 lists the characteristics of the bands, which are in the visible (blue, green, red) and near-infrared part of the spectrum. To illustrate the Sentinel-2 imagery used for analysis, fig. 2 shows an image of the *Zandmotor* indicating Normalized Difference Vegetation Index (NDVI). This ratio takes advantage of the contrasting reflection of photosynthetically active vegetation at visible and near-infrared wavelengths and is widely used for detection and classification of vegetated areas

(Rouse Jr et al., 1974; Tucker, 1979).

**Table 1.** Sentinel-2A bands in VIS and NIR used for linear spectral unmixing procedure

| Band | Central wavelength (nm) | Bandwidth (nm) | Pixel size (m) |
|---|---|---|---|
| 2 - Blue | 496.6 | 98 | 10 |
| 3 - Green | 560.0 | 45 | 10 |
| 4 - Red | 664.5 | 38 | 10 |
| 8 - NIR | 835.1 | 145 | 10 |

### 2.3.1 Linear spectral unmixing

The four selected bands were stacked into a new multispectral data cube and a linear spectral unmixing procedure was applied. This was done to derive sub-pixel proportions of dune cover by marram grass, the most prominent and abundant dune-building species. Linear spectral unmixing is an approach to determine the relative abundance of user-specified ground cover compo-

10 nents (endmembers) in multispectral (or hyperspectral) imagery based on its spectral characteristics (e.g., Smith et al., 1985; Settle and Drake, 1993; Theseira et al., 2003). The reflectance at each pixel of the image is assumed to be a linear combination of the reflectance of the endmembers present within the pixel:

$$R_i = \sum_{k=1}^{n} (f_k \, R_{ik}) + e_i \tag{1}$$

Where $i = 1, ..., m$ are the number of spectral bands, $R_i$ is the reflectance of band $i$ of each pixel, $k = 1, ..., n$ are the number

of endmembers, $f_k$ is the proportion of endmember $k$ within each pixel, $R_{ik}$ the spectral reflectance of endmember $k$ within each pixel on band $i$, and $e_i$ is the residual error term (Lu et al., 2003). Here, two endmembers were specified (see Fig. 2). The first endmember was made up by a group of pixels ($\sim 8$) containing only beach sand, the second endmember by a similarly sized group of pixels fully covered by marram grass. The spectra of the two endmembers were obtained for each Sentinel-2 image separately, and maps containing sub-pixel proportions of beach sand and marram grass were derived using ENVI version

4.8 (Exelis Visual Information Solutions, Boulder, Colorado). The sub-pixel proportions of marram grass were subsequently interpreted as a percentage dune cover within each 10 meter pixel

The linear spectral unmixing procedure was validated against a high-resolution orthophoto of a strecth of foredune directly adjacent to the *Zandmotor* (see also fig. 3). The georeferenced orthophoto (5 cm pixel size) was obtained by an Unmanned Aerial Vehicle (UAV) during a flight on 01-09-2016, so 10 days before the acquisition date of the 2016 Sentinel-2 image. Using





a *k*-means clustering algorithm (Hartigan and Wong, 1979), individual pixels were classified either as beach sand or marram grass. The accuracy of the algorithm was confirmed by visual inspection. The 5 cm pixels were subsequently aggregated into individual 10 m pixels and dune cover was calculated as the proportion of pixels classified as marram grass within each aggregated 10 m pixel.

*FIGURE 2 ABOUT HERE.*

## 3 Results

Figure 3 shows the results of validating the linear spectral unmixing procedure on the Sentinel-2 images. The dune cover calculated from the orthophoto and the two Sentinel-2 images are plotted against each other in fig. 3B. It is clear that deriving sub-pixel proportions of dune cover using linear spectral unmixing result in an overestimation of dune cover by marram grass.
Even though 54% of the variance for 2016 Sentinel-2 image can be explained by a positive linear regression model, most of the data points deviate from the 1:1 identity line because of higher dune cover values calculated by the Sentinel-2 image. This trend, however, appears to be systematic to the linear spectral unmixing procedure since the data points from the 2017 Sentinel-2 image deviate even further from the identity line. This lower correlation ($R^2 \approx 0.35$) is in line with expectation as dune cover by marram grass was observed to have increased at this location between 2016 and 2017. So even though the linear
spectral unmixing procedure overestimates the sub-pixel proportions of dune cover, the derived marram grass cover values for each Sentinel-2 image appear to be comparable relative to each other.

*FIGURE 3 ABOUT HERE.*

Figure 4 shows the derived maps used to identify favorable accommodation space for dune development. The considered Delfland coast domain covers an subaerial area of about 500 ha and about 75% of the coastline (shaded in gray) has been
reinforced with sand nourishments between 2009 and 2011. The first map (4A) gives an overview of the morphological features, including the *Zandmotor*, during early spring 2017. The beach ranges between 0 - 6 m +MSL in height and this is where new embryo dunes have either formed or expanded since 2011. The foredunes stretch fully along the coastline, albeit at variable widths, and are situated at heights between 6 - 14 m +MSL. This indicates that the foredunes, since their construction in 2011, have been raised in height by 2 meter due to aeolian deposition. Older established dunes are excluded from the analysis as
they are minimally exposed to marine forces and mostly covered with vegetation species other than marram grass. All man-made structures on the beach related to coastal safety (e.g. groynes) and leisure and recreation are also excluded. The second map (4B) shows how the subaerial coastal morphology changed between 2013 and 2017, expressed by the average yearly change in height (m/y). It is clear that the most seaward part of the *Zandmotor* experienced strong erosion due to marine forcing, while most of the foredunes and particularly the beach just south and north of the *Zandmotor* experienced accretion
due sand spreading effects. These morphological dynamics are more distinctly reflected in graph 4C, which shows the average alongshore change in coastal sand volumes (m³/m/y) between 2013 and 2017. On average the Delfland coastline has been



accretive, at a rate of 19.3 m³/m/y, but it is clear that there has been a high alongshore variability in sand accretion and erosion rates. This can be attributed to the anticipated behavior of the *Zandmotor*: the accretive areas on its flanks gained approximately 2.8 10⁵ m³/y of sand, while the erosive areas on its base lost about 1.7 10⁵ m³/y of sand. Further, the foredunes experienced accretion of sand at relatively stable alongshore rates. In total, at an average rate of 11 m³/m/y, the foredunes along the Delfland

coast gained approximately 1.6 10⁵ m³/y of sand between 2013 and 2017.

The third map (4D) shows how the dune cover by Marram grass (expressed as percentage per 10 meter pixel) changed along the Delfland coast between the acquisition dates of the two Sentinel-2 images. Using changes in marram grass cover as a proxy for dune development, it appears that in most places the dunes along the coastline have been expanding over the course of a year. This observation, however, must be considered with some reservation, as observed changes in marram grass cover may

also have been due to denser or taller growth of marram grass and not because of actual lateral expansion. Having said that, map 4D suggests that particularly the embryo dunes have been expanding, from 3 to 5 ha between 2016 and 2017. As a result, in 2017 about 17% percent of the dunes along the Delfland coast could be considered embryo dunes, of which most developed naturally along the coastline. The foredunes, in contrast, appear to have experienced more spatial variability in marram grass cover changes. Map 4D suggests that, along most of the Delfland coast, the foredunes have been expanding between 2016 and

2017. However, especially at the dune toe and just leeward of the dune crest, the foredunes appear to have declined somewhat in cover. This decline is most apparent north of the *Zandmotor*, which is clearly reflected in the second graph (4E) that shows the alongshore yearly change in dune cover (m²/m/y) between 2016 and 2017. This northerly foredune decline will be examined in more detail in the discussion, but it can likely be attributed to anthropogenic disturbances (due to recreational activities as well as nature management practices) and to the fact that this stretch of coastline has not been nourished with sand when the

Delfland coast was reinforced between 2009 and 2011. All in all, data from Sentinel-2 imagery suggests that the dunes along the Delfland coast have been expanding between 2016 and 2017 at an average rate of about 11.2 m²/m/y. For the foredunes this amounted to an increase of dune cover of 42 to 54 ha. Though, as stated before, this may be an exaggerated number as increase in dune cover by marram grass has likely not been due to lateral growth alone.

## *FIGURE 4 ABOUT HERE.*

Accommodation space is considered favorable for dune development when it is (1) sheltered from storm impact and (2) experiencing a steady accumulation of wind-blown sand. The first (boundary) condition is identified by comparing the presence of dunes in 2017 to the height at which they were located. As fig. 5A demonstrates, in 2017 there were no dunes present below a height of 1.6 m +MSL. All embryo dunes were located on the beach between 1.6 - 6 m +MSL, while the foredunes were located at heights between 6 - 14 m +MSL. This suggests that, in 2017, dunes along the Delfland coast were sheltered from

storm impact above a height of 1.6 m +MSL. Accommodation space, as a result, could be considered favorable for dune development above this boundary height. The second condition is identified (or verified) by comparing the changes in dune cover by marram grass from 2016 to 2017 to the average yearly change in dune height between 2013 – 2017 (see fig. 5B). Several observations can be drawn from that comparison but, most importantly, it demonstrates that 95% of all dunes along the Delfland coast in 2017 (both embryo dunes and foredunes) were present in areas that experienced (on average) a continuous





accretion of sand. This indicates that favorable accommodation space for dune development can indeed be characterized by a steady accumulation of sand. And since all dunes in 2017 were located above a height of 1.6 m +MSL, it is reasonable to assume that this accumulation of sand occurred predominantly by aeolian forcing. In addition, fig. 5B shows that almost all dunes increased in cover by marram grass between 2016 and 2017. Overall this increase in cover was most pronounced for the

5 embryo dunes, as the foredunes showed limited increase and even some decrease in dune cover towards higher changes in dune height. The largest increase in dune cover between 2016 and 2017, however, coincided with the same change in dune height ($\sim 0.1$ m/yr) for both the embryo dunes and the foredunes.

## *FIGURE 5 ABOUT HERE.*

The identified accommodation space favorable for dune development (i.e. located above 1.6 m +MSL in height and expe-

10 riencing a steady accumulation of wind-blown sand) is shown in map 6A. Including the parts that were already covered by marram grass, it is clear that large areas along the Delfland coast were favorable for dune development in 2017. Especially the high and accreting southern and middle part of the *Zandmotor* stood out for its large favorable accommodation space for dunes to develop. This is reflected more clearly by graph 6C, which shows the favorable accommodation space along the Delfland coast (in m$^2$/m) as well as the potential for new dune development. As map 6B shows, this potential is calculated by

15 subtracting the dune cover already present in 2017 from the total favorable accommodation space. Graph 6C makes clear that dune development potential along the Delfland coast is mainly reserved to embryo dune development, as most of the favorable accommodation space is located on the beach. Further, graph 6C highlights the overall importance of the *Zandmotor* for dune development along the Delfland coast: By providing the largest favorable accommodation space the *Zandmotor* supports the highest potential for new dunes to develop. Most of this potential is also allocated to embryo dune development, as most of

20 the favorable areas are located on the beach. The existing foredunes show a limited development potential as they are already quite densely covered by marram grass. Still, of the considered 500 ha domain of the Delfland coast, in 2017 an estimated two thirds ($\sim 165$ ha) appears to have provided favorable accommodation space for dune development.

## *FIGURE 6 ABOUT HERE.*

## 4 Discussion

This paper examined to what extent the Zandmotor has contributed to creating accommodation space favorable for dune development. The results indicate that the *Zandmotor* provides the most favorable accommodation space along the Delfland coast, for it has large areas located above 1.6 m +MSL that experience (on average) a continuous accretion by wind-blown sand. As such, the results highlight that the *Zandmotor* supports an especially high potential for new embryo dunes to develop as most of its favorable accommodation space is located on the beach. This section examines the merit of the identified conditions for

when accommodation space is considered favorable for dune development, as well as the merit of using favorable accommodation space to indicate the potential for dune development. The latter is examined in relation to the intended dynamical nature of





the *Zandmotor* and the persistent anthropogenic disturbances arising from recreation and leisure as well as nature management practices.

## 4.1 Boundary conditions indicating favorable accommodation space

Accommodation space is considered favorable for dune development when it is sheltered from storm impact and experiences a steady accumulation of wind-blown sand. The latter condition is not disputed, as the reinforcing feedback between the growth response of marram grass and burial by wind-blown sand is well documented (Huiskes, 1979; Disraeli, 1984; Maun and Lapierre, 1984; Van der Putten et al., 1988; Hesp, 1991; Maun, 1998) and recognized to be fundamental to coastal dune development in temperate regions around the world (e.g., Baas and Nield, 2010; Durán and Moore, 2013; Keijsers et al., 2016; Nolet et al., 2017). The positive feedback mechanism is thought to originate from a trait that all beach grasses of the genus *Ammophila* possess, namely potentially unlimited horizontal and vertical growth through its rhizomes (Gemmell et al., 1953; Ranwell, 1972). Whether marram grass grows horizontally or vertically depends on the amount of wind-blown sand, which is particularly advantageous to dune-building. After establishment on the beach, by seed or rhizome dispersal, marram grass first produces leafy shoots along newly developing horizontal rhizomes. When wind-blown sand is trapped by the leafy shoots, the immediate sand surface is raised and a small embryo dune is formed (Hesp, 1989). The leafy shoots are capable of growing up through a moderate thickness of sand by elongation of individual leaves. If, however, a leafy shoot is overwhelmed by sand burial, one or more of its axillary buds develop into a vertical rhizome that will continue to grow until the surface is reached. Adventitious roots are produced from the nodes of the vertical rhizome and the horizontal rhizomes gradually die, so that the vertical rhizomes become independent of one another. This process may be repeated as long as aeolian supply is abundant and marram grass continues to trap sand. The capacity to trap sand, as noted before, is enhanced by the growth response of marram grass to trapping, which introduces the positive feedback mechanism driving coastal dune development (Gemmell et al., 1953; Ranwell, 1972). Using very high-resolution data, Nolet et al. (2017) showed that marram grass on foredunes along the *Zandmotor* thrives best under a sand trapping rate of approximately 0.3 meter of sand per growing season and that marram grass can withstand sand burial up to 1 meter of sand. However, while this demonstrates how positive plant-sand feedback steers dune development, it must be noted that the physical size of a developing dune also controls its morphology. As dunes grow, for example, a limit is imposed on its height because the wind force required to transport sand upslope increases significantly (e.g., Arens et al., 1995; Arens, 1996; Keijsers et al., 2015a). Established foredunes, as a result, tend to expand in width rather than height, which emphasizes the importance of the wide favorable accommodation space that the *Zandmotor* provides for foredune development.

The condition that accommodation space is considered favorable when it is sheltered from storm impact warrants closer inspection, because the impact depends both on the magnitude of the storm as well as the geometry of the beach (Houser et al., 2008). Wind stress due to atmospheric pressure differences drive storm surge levels and offshore wave conditions, but the vertical dimension of the beach profile, in particular, exerts great control on shoreline parameters such as wave setup, swash and runup (e.g., Stockdon et al., 2006; Sallenger Jr, 2000; Ruggiero et al., 2001). This is significant because the dissipation of kinetic energy of breaking waves is responsible for the highest rates of coastal erosion (e.g., Vellinga, 1982; Short and Hesp,





1982). However, while empirical models can calculate wave runup levels and wave breaking energy from parameters such as offshore wave conditions and beach profile (see Stockdon et al. (2006) and Sallenger Jr (2000) for details), those relations only return approximations as often not all required model input is available or because of inherent model uncertainties. Having said that, the results suggest that dunes along the Delfland coast are sheltered from storm impact above a beach height of

1.6 m +MSL. This finding is examined in relation to offshore sea water levels measured by a buoy in close proximity to the *Zandmotor* mega-scale beach nourishment. Figure 7 shows the probability density curve (which is bimodal because of tidal dynamics) of those sea water levels (in m ± MSL), measured every 10 minutes from 2011 until 2017. Included are the instances when sea water levels exceeded the apparent 1.6 m +MSL boundary height for dunes to be sheltered from storm impact. It is clear from fig. 7 that this did not occur frequently, only during about 0.4% of the measurements. Those measurements, however,

were relatively clustered together, meaning that the boundary height was exceeded over (relatively) prolonged periods of time. Although, over the course of six years this happened for no more than 10 full days. On average the exceedance was about 20 cm up to a sea water level of 1.8 m +MSL, but on a few occasions sea water levels almost doubled compared the boundary height to 3.10 m +MSL. This is excluding the wave runup onto the beach, which can be significant for natural beaches in the Netherlands. Dependent on whether the beach profile is dissipative or reflective, both Stockdon et al. (2006) and Poortinga

et al. (2015) show that wave runup may reach to heights from 0.85 to 1.45 m above still water level (m), which is the level that would occur in the absence of waves. This implies that, since the construction of the *Zandmotor* in 2011, the Delfland coast may have experienced coastal erosion by storm surge levels that reached to heights of at least 4 m +MSL.

The observation that in 2017 quite a large number of embryo dunes were present on the beach at heights well below the maximum experienced storm surge levels, points to the capacity of dunes to withstand and recover from hydrodynamic storm

impact and the pivotal role marram grass plays herein. In essence, as remarked by various researchers (e.g., Suanez et al., 2012; Feagin et al., 2015; Houser et al., 2015; Van Puijenbroek et al., 2017a, b), the ability of embryo dunes to recover from storm impact largely depends to what the extent the above- and belowground structural integrity of marram grass remains intact after a storm event. This depends, in turn, on the severity of the storm impact on the dune, which can be caused by wave erosion (scarping and overwash) and swash inundation (Sallenger Jr, 2000; Hesp and Martínez, 2007). Wave erosion may completely

remove all sand from an embryo dune (so it is no longer raised from the beach surface) and have an abrasive effect on the leaves of marram grass, causing either minor damage or compete removal of all aboveground biomass. Most of the belowground root system of marram grass, however, has been observed to largely remain intact after wave scarping or overwash (Feagin et al., 2015). Potential damage of swash inundation on marram grass depends on the duration of the inundation, but as Vergiev et al. (2013) demonstrate, marram grass displays no visible decomposition of stems, roots or rhizomes after being immersed with sea

water for 20 days. This is well beyond the period a beach will be inundated after a storm event, which implies that inundation has a limited, if any, negative effect on the structural integrity of marram grass. Given that storm events occur more frequently in winter, it has been observed that embryo dunes on dissipative beaches undergo a classic seasonal cycle of erosion during the winter and accretion during the summer (Montreuil et al., 2013; Van Puijenbroek et al., 2017b). Their presence on the beach, however, would remain persistent throughout the year and often show a yearly net growth when aeolian supply was

sufficient (Anthony et al., 2007; Suanez et al., 2012). This not only indicates that embryo dunes have the capacity to withstand



storm impact and quickly recover to prestorm conditions, but also that this is most likely because the above- and belowground structure of marram grass often remains largely intact after a storm event.

## *FIGURE 7 ABOUT HERE.*

### 4.2 Dune development potential in relation to anthropogenic impacts

The results highlight the overall importance of the *Zandmotor* for dune development along the Delfland coast. First, this is because its beach provides very wide favorable accommodation space that therefore supports a high potential for new embryo dune development. And second, because of its sand feeding effects, the *Zandmotor* has likely contributed to creating more favorable accommodation space for dune development along the entire Delfland coast. The coastline directly north of the *Zandmotor*, for example, experienced a significant accumulation of sand between 2013 and 2017 even though it has not been

nourished with sand in the years before. Although the amount of sand accumulation was less compared to the coastline that has been nourished between 2009 and 2011, the overall positive sand budget illustrates the intended dynamical nature of the *Zandmotor*, where its sand is redistributed along the coastline causing a seaward broadening of the beach and dunes. In fact, graph 6C suggests that the unnourished northern part of the Delfland coastline supports a higher potential for dune development compared to the nourished southern coastline. In part this may be due to the fact that the Delfland coast is characterized by a

net northward sediment transport regime (Van Rijn, 1997), which is reflected in the sand feeding budget of the *Zandmotor*. In the first 18 months after its completion, De Schipper et al. (2016) for example show that up to 40% more sand of *Zandmotor* was transported in a northward direction rather than southward towards Rotterdam harbor. At the same time, because the 2009 – 2011 nourishment strategy consisted (for a large part) of foredune reconstruction that included plantings of marram grass, the created favorable accommodation space along the nourished coastline may not be accommodating to much new dune

development. As such, even though the coastline south of the *Zandmotor* has been reinforced with sand nourishments, it is quite possible for the unnourished northern Delfland coastline to experience more pronounced dune development in the years to come.

Interestingly, however, the positive effect of the *Zandmotor* on the northern Delfland coastline, in terms of sand accretion, is not reflected in the changes of cover by marram grass between 2016 and 2017. Even though it is shown that the coastline

north of the *Zandmotor* provides ample favorable accommodation space, it appears that the potential for dune development is currently not being realized. There are two main anthropogenic impacts that may hamper dune development along this urbanized coastline, namely persistent disturbances arising from recreation and leisure as well as a (increasingly prevalent) nature management practice that is aimed at remobilizing the dune landscape. Figure 8 gives an overview of total alongshore changes in dune cover by marram grass between 2016 and 2017 (in $m^2$/m/y) and aims to relate it to anthropogenic activities that

may impact (both positively and negatively) dune development along the Delfland coast. In the broadest sense, the coastline can be divided first according to whether or not it has recently been nourished with sand and what type of sand nourishment has been implemented. A distinction can be made between the sand nourishments that were carried out between 2009 and 2011 to reinforce most of the Delfland coastline, the nature compensation project *Spanjaardsduin* implemented at the same time and,



finally, the *Zandmotor* mega-scale beach nourishment that was completed at the end of 2011. Further, within the nourished coastline there is a zone where the dune development appears to lag behind compared to the rest of the nourished coastline. This zone, as fig. 8B illustrates, can be characterized by a higher concentration of disturbances arising from recreation and leisure. Then, finally, there is the northern part of the Delfland coastline has not been nourished with sand in recent years. Within this

zone, as fig. 8C shows, relatively large dune areas have been excavated between 2011 and 2013 aimed at rejuvenating the dune landscape by reinitiating aeolian dynamics. The following paragraphs discusses each identified zone of the Defland coast and how dune development potential may be impacted by the various anthropogenic activities.

As fig. 8C shows, the overall positive effect of the three sand nourishment schemes on dune development is clearly reflected in the changes of marram grass cover between 2016 and 2017. Within the nourished zone, however, there are three clear dips

where the dune cover appears to have decreased over the course of a year. Upon closer inspection it seems that each dip coincides with a beach entrance where the public can enter the beach. A number of natural processes and human activities may be involved here in the observed decline in marram grass cover. First of all, as fig. 8B shows, the seaward side of a beach entrance is commonly paved with concrete slabs and cuts relatively deep into the stoss slope of the foredune. This, effectively, mimics a through foredune blowout (e.g., Hesp, 2002), in which wind erosion is enhanced because of local wind

speed acceleration and pronounced turbulent flow structures such as corkscrew vortices (Hesp and Martínez, 2007). Because the floor is paved, these wind-driven forces will in particular erode (i.e. widen) the slopes of the beach entrance and this susceptibility to lateral erosion may have lead to the observed decline in marram grass cover. Second, as can also be seen in fig. 8B, there is often a hospitality establishment (e.g. a beach bar or restaurant) directly besides a beach entrance. And although their placement on the beach is often seasonal, their presence is numerous. In the summer of 2017, for example, only three of

the twenty-three beach entrances along the Delfland coast did not have one or more hospitality establishments directly placed besides it. Perhaps not coincidentally, two of those three entrances gave access to the more isolated parts of the *Zandmotor*. The presence of hospitality establishments puts additional pressure on the dunes as people may flock around the beach entrances and motorized vehicles are more common, for example to resupply the establishment. Even though walking or driving in the foredunes is prohibited along the Delfland coast, several studies (e.g., Andersen, 1995; Anders and Leatherman, 1987) show

that vehicles and people on the beach may have a significant negative effect on dune development.

At the same time, as laid out in more detail by Jackson and Nordstrom (2011), the structure of the hospitality establishment itself may alter (i.e. block) aeolian transport from the beach and retard foredune development. This plays an even larger role in the nourished zone north of the *Zandmotor* where, besides a large number of hospitality establishments, also a high amount of seasonal beach cabins are placed along the toe of the foredunes from March till October (from 2016 onward). Their placement

close together, as fig. 8A shows, has the additional effect that the airflow can constrict and accelerate between the beach cabins, which increases the likelihood of local scour resulting in aeolian deposition farther landward (Nordstrom, 2004). Although in some instances this may actually be beneficial to the foredunes, the clear decline in cover by marram grass (see fig. 8C) indicates that the placement of beach cabins had an overall negative impact on dune development between 2016 and 2017. And while the beach cabins were in fact raised slightly from the surface ($\sim$ 50 cm), Nordstrom and McCluskey (1984) show that

such a modest height may not have been sufficient to minimize interference with the wind flow and resultant aeolian dynamics.



Further, another important anthropogenic disturbance with a highly negative impact to dune development, is that the beach directly north of the *Zandmotor* is mechanically raked during the summer to remove wrack line material and human litter. Even though it is a common practice to accommodate beach recreation (Jackson and Nordstrom, 2011), this severely hampers embryo dunes from establishing themselves on the beach. Not only can the used machinery destroy any sprouting seedlings or rhizomes of marram grass, the removal of wrack deposits also deprives marram grass from potential hospitable locations to establish itself on the beach (Kelly, 2014). As a result, these anthropogenic disturbances combined have likely contributed to reduced dune development compared to the rest of the nourished Delfland coastline.

Then, as fig. 8C and fig. 4E as show, the decrease of marram grass cover along the unnourished northern part of the Delfland coastline suggests that the foredunes have been in decline between 2016 and 2017. Which is unexpected considering the positive sand feeding effect of the *Zandmotor* on this stretch of coastline. Upon closer inspection, the main candidates for the observed foredune decline are a number of dune excavations aimed at rejuvenating the dune landscape. When the focus of Dutch coastal policy widened, to also include preserving the spatial quality and natural values of the coastal zone, it was recognized that traditional flood safety measures had led to over-stabilized dune systems that were characterized by a markedly reduced biodiversity compared to younger and more dynamic dune systems (e.g., van Dorp et al., 1985; Provoost et al., 2011). For that reason, in places where coastal safety could be guaranteed, remobilizing dune systems by removal of dune vegetation and topsoil has become a key management practice for maintaining a high biodiversity in the dune landscape. Reinitiating aeolian dynamics is hereto essential, as deflation and deposition zones creates habitat diversity and renewed opportunities for specialized pioneer vegetation species (e.g., Arens et al., 2013). Nowadays, in order to maintain or even increase dune mobility, the rejuvenated dune systems are often connected to the beach and foredunes through the excavation of foredune notches. This has been shown to result in a sustained input of wind-blown calcareous beach sand and more diverse living conditions for pioneer vegetation, e.g due to higher levels of sand burial, wind speeds or salt spray (Riksen et al., 2016; Ruessink et al., 2017). However, as fig. 8C shows, dune excavation practices appear to adversely affect foredune development. While there were no foredune notches explicitly excavated along the Delfland coast, fig. 8B shows that the paved beach entrances may similarly act as conduits for aeolian transport into the dune excavations. As a result it is quite possible that the narrow foredunes in front of the excavated dune are experiencing a net deflation of sand which negatively affects the growth of marram grass (e.g. by root exposure). Sand deposition then likely occurs deeper landward where no marram grass is presently growing to benefit from an increase of sand burial. Marram grass cover, as a result, has been in decline between 2016 and 2017, indicating a decline in the foredunes as well. However, with the *Zandmotor* feeding the coastline and providing an effective flood defense, this localized foredune decline should not pose an imminent threat to coastal safety.

*FIGURE 8 ABOUT HERE.*



## 5 Conclusions

This paper examined to what extent the *Zandmotor* has contributed to creating accommodation space along the Delfland coast favorable for dune development, i.e. accommodation space that is sheltered from storm impact and experiencing a steady accumulation of wind-blown sand. Comparing the presence of dunes in 2017 to its elevation indicates that dunes are sheltered from storm surges above a height of 1.6 m +MSL. Comparing the changes in dune cover by marram grass from 2016 to 2017 to the average yearly change in dune height between 2013 – 2017 demonstrates that dunes were almost exclusively present in accreting areas. As such, the results highlight the overall importance of the *Zandmotor* to dune development potential:

– Compared to the rest of the Delfland coast, the supratidal beach of the *Zandmotor* provides very wide favorable accommodation space and therefore supports a high potential for new embryo dunes to develop.

– Because of its sand feeding effects, the *Zandmotor* will likely contribute to creating more favorable accommodation space for dune development along the entire Delfland coast.

However, because of persistent anthropogenic disturbances arising from recreation, leisure and nature management, dune development along this urbanized coastline may not reach its full potential. This should not be too alarming, though, as the *Zandmotor* mega-scale beach nourishment is set to ensure the safety of the Delfland coast for years to come.

*Data availability.* Coastal lidar data is made available under a Creative Commons Zero (CC0) statement by Rijkswaterstaat, the Dutch executive agency of the Ministry of Infrastructure and Water Management. Sentinel-2 satellite imagery is made available by the European Space Agency (ESA) in agreement with the Copernicus Sentinel data policy.

*Competing interests.* The authors declare no competing interests

*Acknowledgements.* The research was carried out within the program Nature-driven nourishment of coastal systems (NatureCoast) and funded by technology foundation STW (grant 12686), applied science division of the Netherlands Organization for Scientific Research (NWO).



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



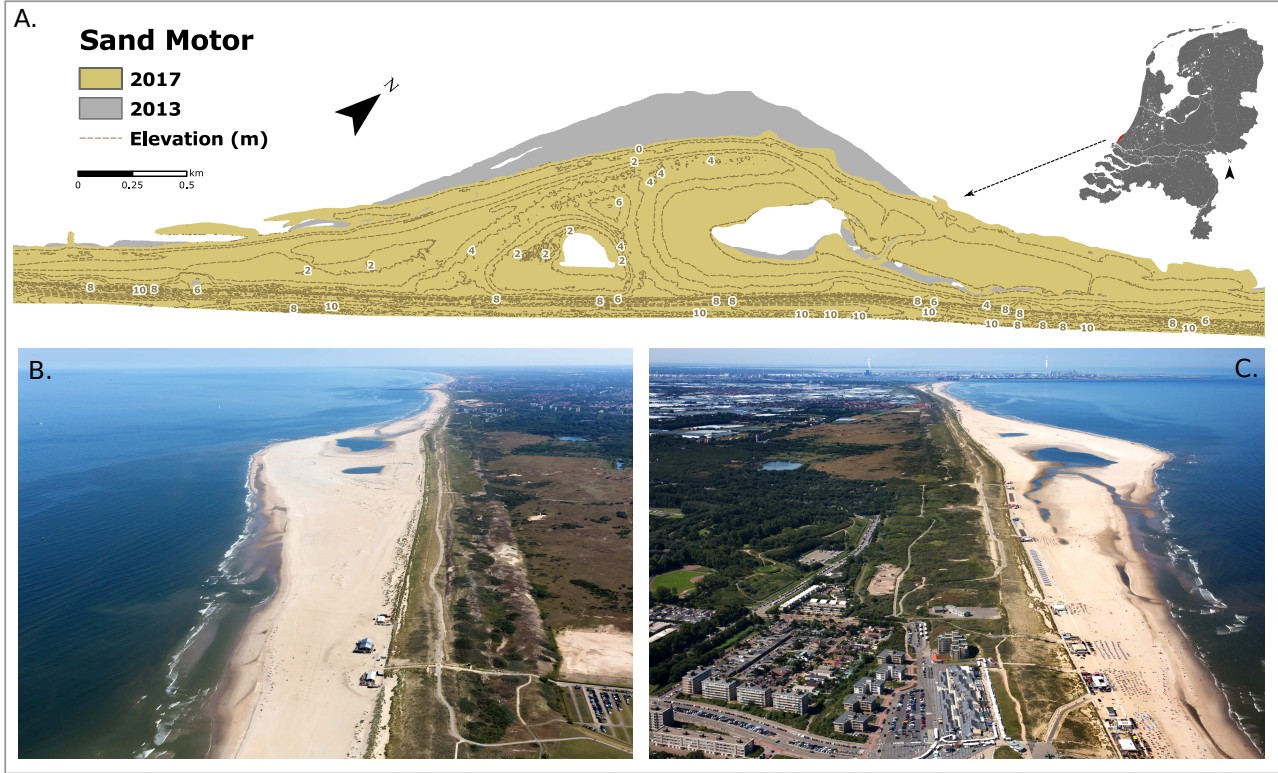

**Figure 1.** The *Zandmotor* (Sand Motor), an experimental mega-scale beach nourishment located along the Delfland coast, the Netherlands. **1A**. The design of the *Zandmotor* mirrors the onshore migration of an (intertidal) sandbar and aims to provide coastal safety by redistributing its sand along the coastline, thereby broadening the beach and dunes. **1B-C**. Aerial photographs of the *Zandmotor* and Delfland coast taken on July 9 2017, facing north (B) towards The Hague and south (C) towards the harbor of Rotterdam. The Delfland coast maintains relatively wide and natural dune areas, but in some places the dunes are not more than a narrow foredune ridge directly bordered by urbanized areas.





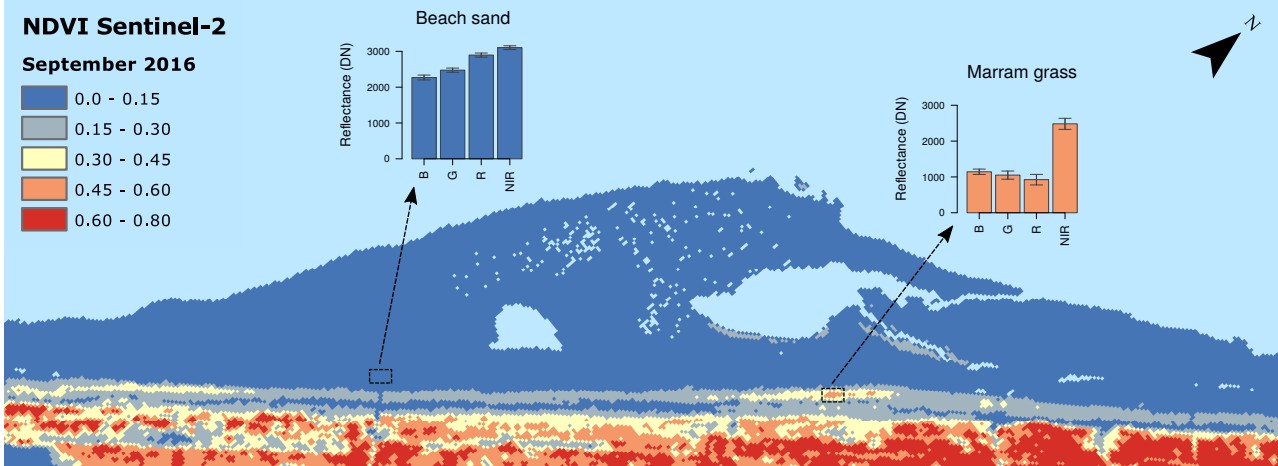

**Figure 2.** Map of the *Zandmotor* indicating Normalized Difference Vegetation Index (NDVI), to illustrate the Sentinel-2 imagery and selection of endmembers (user-specified ground cover components) for the linear spectral unmixing procedure. Sub-pixel proportions of beach sand and marram grass (for every 10 m pixel) were derived using endmember reflectance spectra for four bands in the visible and near-infrared spectrum.



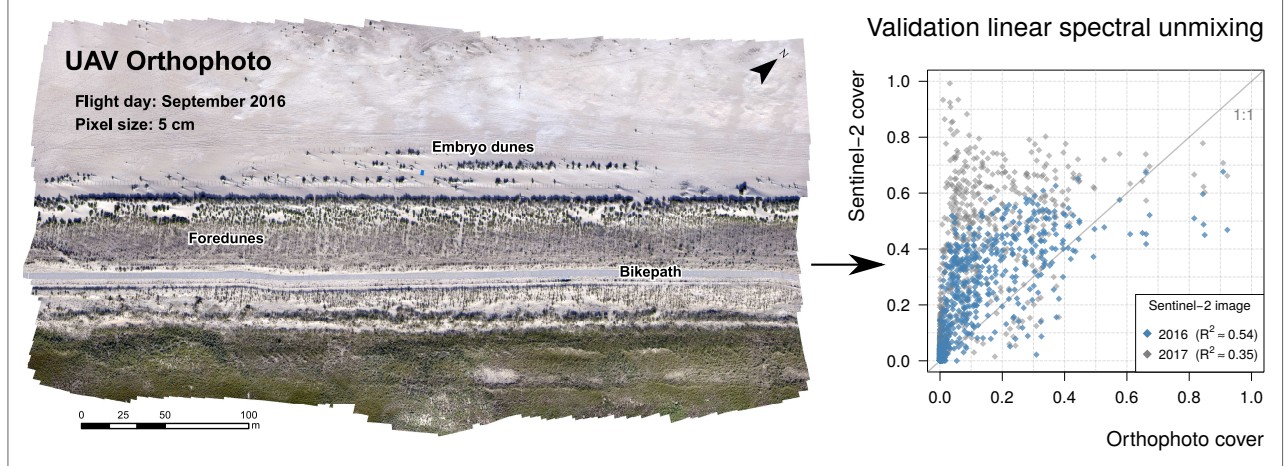

**Figure 3.** Validating the linear spectral unmixing procedure on the Sentinel-2 images using an orthophoto (5 cm pixel size) derived in September 2016 by an Unmanned Aerial Vehicle (UAV). Linear spectral unmixing produces a systematic overestimation of sub-pixel proportions of marram grass cover compared to the cover values calculated from the orthophoto. A positive linear regression model explains 54% of the variance for the 2016 Sentinel-2 image and 35% of the variance for the 2017 image.





**Figure 4.** Morphological changes and dune vegetation cover dynamics of the Delfland coast. **4A**. Map of the considered domain including the relevant morphological features (beach, foredunes, embryo dunes) in 2017. **4B**. Map of subaerial coastal morphological changes between 2013 and 2017, expressed average change in height per year (m/y). **4C**. Average alongshore changes in coastal sand volumes ($m^3$/m/y) between 2013 and 2017, differentiated by the beach and foredunes. **4D**. Map of changes in dune cover by marram grass between 2016 and 2017, expressed as percentage per 10 m pixel. **4E**. Alongshore changes in marram grass dune cover ($m^2$/m/y) between 2016 and 2017, with differentiation between the foredunes and embryo dunes.



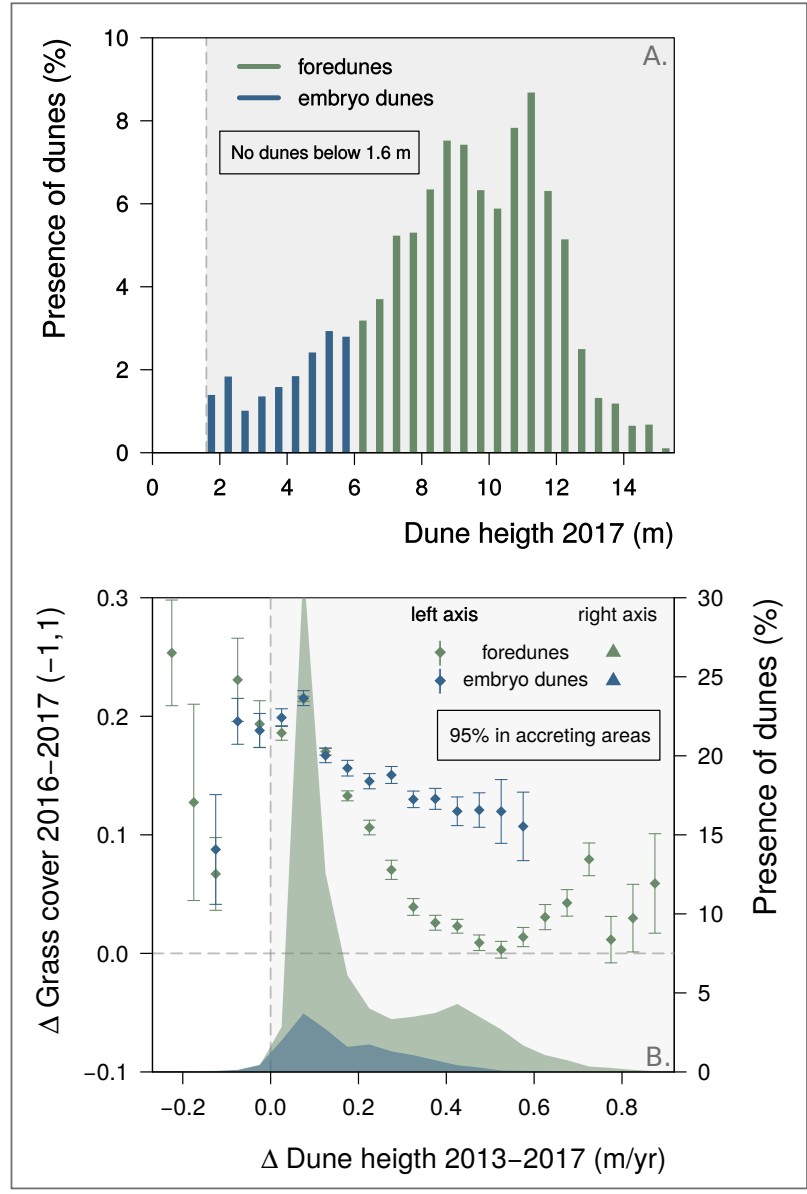

**Figure 5.** Identifying favorable accommodation space for dune development (both foredunes and embryo dunes) along the Delfland coast **5A**. Comparing the presence of dunes in 2017 to its elevation: no dunes present below 1.6 m +MSL. **5B**. Comparing the changes in dune cover by marram grass from 2016 to 2017 to the average yearly change in dune height between 2013 - 2017: most dunes were present in accreting areas (right axis) and almost all dunes increased in cover by marram grass between 2016 and 2017 (left axis).





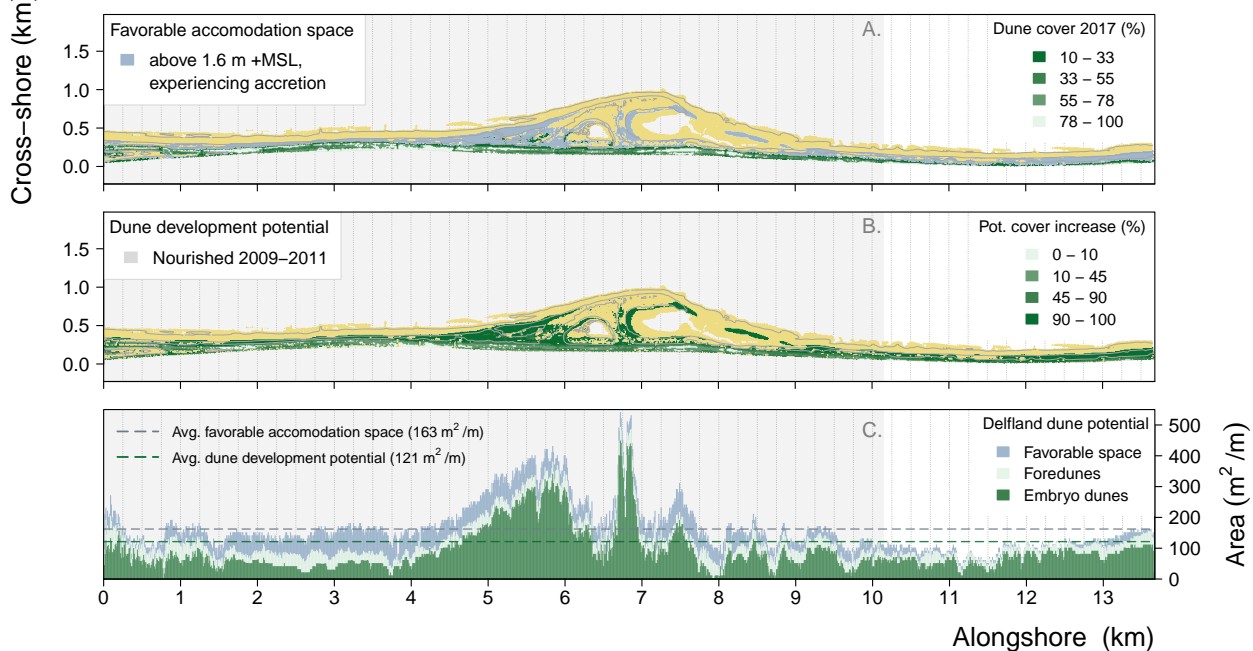

**Figure 6.** Favorable accommodation space to indicate dune development potential along the Delfland coast. **6A**. Map of dune cover (%) by marram grass in 2017 and accommodation space favorable for dune development: located above 1.6 m +MSL and experiencing an accretion of sand. **6B**. Map indicating Delfland coast dune development potential, calculated as the potential of marram grass to increase in cover (%) based on the accommodation space favorable for dune development and taking into account the dune cover already present in 2017. **6C**. Alongshore variation of favorable accommodation space and potential for dune development along the Delfland coast (m²/m), differentiated for foredunes and embryo dunes.





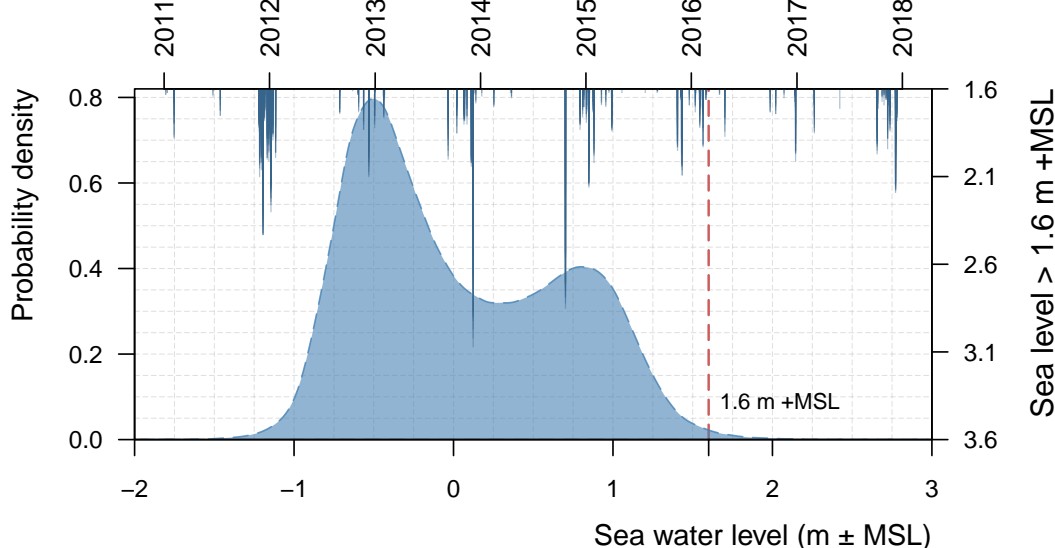

**Figure 7.** Bimodal probability density curve of sea water levels (in m ± MSL) measured by a buoy near Scheveningen since the construction of the *Zandmotor* in 2011 until 2017. Included are the instances when sea water level exceeded the 1.6 m +MSL boundary height for dune development along the Delfland coast.



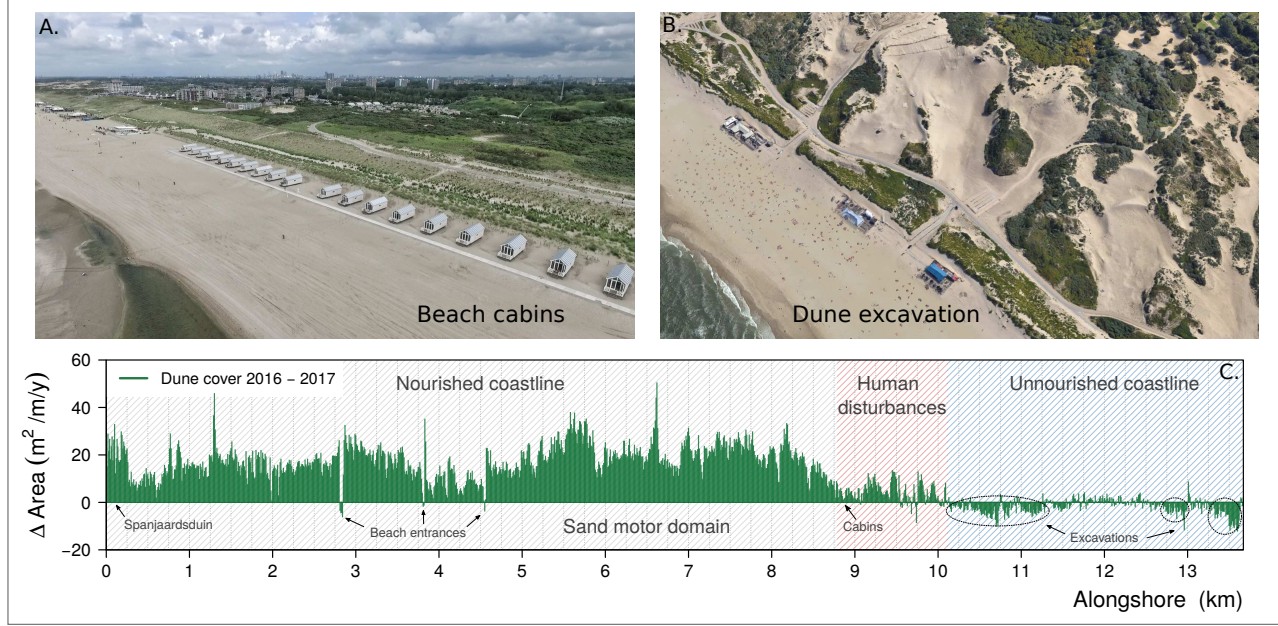

**Figure 8.** overview of total alongshore changes in dune cover by marram grass between 2016 and 2017 (in m$^2$/m/y), in relation to anthropogenic activities that impact (positively and negatively) dune development along the Delfland coast. Aerial photographs courtesy of René Oudshoorn (**8A**) and Google Maps (**8B**).