# Peer review of "Accommodation space indicates dune development potential along an urbanized and frequently nourished coastline"

_Earth Surface Dynamics, 2018_

## Referee Comment (RC1) · M. Hilton (Referee) · 21 Jul 2018

Comments on paper by Nolet and Ricksen submitted to Earth Surface Dynamics Mike Hilton, University of Otago michael.hilton@otago.ac.nz

Always fascinating to read about the efficacy of the Zandmotor and I'm sure there will be a wide audience for this paper. Specifically the authors consider the potential for this intervention to create space for the growth of marram grass and the development of embryo dunes.

They approach this question primarily from an earth science perspective. They identify

the 'beach' above a height subject to storm surge and which has experienced sand accretion. Incidentally, I find it difficult to apply the term 'beach' in this context. Also, a 12-month period is a very short period to assess embryo foredune development. Was this a typical 12-month period in relation to storm surge and wind stress?

The results point to the establishment of a large accommodation space as a result of the nourishment. Foredunes have developed adjacent to the Zandmotor, presumably (and forgive me if the authors explain this) as a result of bioengineering. The development of embryonic dune across the beach, since culmination of the Zandmotor, has been patchy. Even the embryonic dunes at the back of the beach, adjacent to Duinfietspad (the road boarding Zandmotor) – some of which have been established from deliberate plantings of marram - have barely flourished.

There is an alternative – ecological - perspective on this situation. A considerable section of the paper is committed to explaining how marram grass traps sand and grows by the vertical and horizontal extension of rhizome. But what the germination and growth requirements of this species? Does the surface of the Zandmotor, or most of the surface, satisfy these requirements? Further, marram growth and embryo dune development results from the deposition and germination of marram grass seed and / or rhizome. The process of marram rhizome dispersal and growth is well documented in these papers:

Konlechner, T. M., Hilton, M. J., & Orlovich, D. A. (2013). Accommodation space limits plant invasion: Ammophila arenaria survival on New Zealand beaches. Journal of Coastal Conservation, 17(3), 463-472. doi: 10.1007/s11852-013-0244-5.

Konlechner, T. M., Orlovich, D. A., & Hilton, M. J. (2016). Restrictions in the sprouting ability of an invasive coastal plant, Ammophila arenaria, from fragmented rhizomes. Plant Ecology, 217(5), 521-532. doi: 10.1007/s11258-016-0597-6.

In short, if marram propagules are not being deposited on the surface of the Zandmotor we cannot expect the development of dunes. Moreover, a range of conditions will limit

the germination of seed and the success of growth from rhizome. In the case of the former the seed must remain close to the surface (within 7cm) to avoid dormancy (to germinate and to avoid entering the long-term seedbank). A very active substrate, where there are high rates of sedimentation, may also result in unfavourable growth conditions, even if germination occurs. A wind rose was not included in the paper, but I recall the drift potential is generally to the northeast ... so there is no source of seed upwind of the Zandmotor? Similarly, the likelihood of the waves depositing rhizome on the beach is low or nil because of the elevation of the beach.

Marram growth from either seed or rhizome is inhibited by high moisture content and the surface of the beach appears to be corrugated with alternating transverse protodunes and moist deflation areas. One last (ecological) point. Marram requires nutrients to grow and so growth is usually most vigorous immediately adjacent to a beach, where decomposing algae and marine debris provide abundant fuel for growth. I wonder what the nutrient status of the substrate of the Zandmotor might be? It is probably quite low.

Therefore, if I'm correct, the actual accommodation space generated by the Zandmotor might be a good deal less than you estimate. I would like to see some experimental work on the Zandmotor, with the goal of examining the biological requirements of marram grass and the dispersal of propagules. Maybe the authorities should plant more marram? But where to maximise success? It may not be in the areas experiencing the most rapid accretion! Because in these areas the rate of accretion may exceed the tolerance of seedlings and be too dry for growth from rhizome (which would not be deposited at these sites in any case).

A couple of other issues. I found the maps very difficult to interpret (although I was dealing with the grey-scale versions). I very much enjoyed reading about the coast to the north and the constraints imposed by use of the coast.

Finally, my apologies for the time it has taken me to comment. I value the paper and it is a worth contribution provided you acknowledge you are looking at a discrete element

СЗ

of the (biogeomorphic) dune system.

---

## Referee Comment (RC2) · I. Delgado-Fernandez (Referee) · 27 Jul 2018

Thank you for allowing me to comment on your article and apologies for the delay in sending this review. I hope my feedback below is useful and I will be happy to further discuss changes to the paper should you wish to do so.

I have very much enjoyed your article, and I believe there is good material in it. I think the text needs work though and I would concentrate in two main areas: (1) the methods and (2) the definition of 'favorable' accommodation space. You have a really good dataset but the pixel resolution of both your LiDAR and Sentinel data has limitations. The error introduced by these limitations could be quantified, as this has implications

for both your results/conclusions and for the maps you produce. Additionally, some of the generalizations in this article may need field validation (in my opinion), especially those related to the concept of 'favorable accommodation space'. My concern with this concept (as it is currently defined in the text) is that it does not consider surface conditions having an effect on aeolian transport along this section of the coastline (which could in fact determine whether the accommodation space is indeed 'favorable' or not) or conditions for the colonization of new marram grass seeds on the beach. It might be a matter of further discussing this in the article? (I have included some ideas below which you may want to consider). Finally, and although I can see the benefit of looking at long sections of the coast, I feel that the article has the potential of providing much more detail on the morphological evolution of different areas by, for example, zooming into particular sections of the dunes.

Here are a few major comments:

1. Very nice introduction, and great to read the contextual and historical information about this part of the Dutch coast in section 2.1. I think, however, that there is a need to include some additional information on the peculiarities of aeolian transport at the Zandmotor. The paper gravitates around 2 conditions for the development of coastal dunes: 1) steady supply of wind-blown sand; and 2) wide accommodation space (or simply beach width). The second is important as it is related with fetch distances, etc. (I'm suggesting the paper by Delgado-Fernandez, 2010, on the fetch effect because it is a review article but please choose any other that may suit you well), additional to reducing storm impact as you describe in the article. However, accommodation space is only one of the many variables involved in sediment input to coastal dunes. Aeolian transport is a function of many surface conditions including the formation of lag deposits or the composition of sediment, which seems to be an important limiting aspect at the Zandmotor (as described by the group of de Vries and Honhout, etc.). This is, the Zandmotor does seem to have created accommodation space but the question is how 'favourable' this accommodation space really is for maximizing aeolian transport and

hence dune growth (not necessarily just limiting dune erosion). I would recommend that potential supply-limiting conditions affecting transport by wind at this location are reviewed in the introduction as these affect your condition n. 1 ('steady supply of wind-blown sand').

2. The inclusion of supply limiting conditions above might have implications for your discussion. Your section 4.1 suggests that you are using the presence of vegetation as a proxy for sediment input to the dunes (something I come back to in comment n. 9). I do not think 1 year is long enough for the analyses you make. However, even if it was, your figure 4 seems to suggest that vegetation in areas other than the Zandmotor has followed a similar trend than those fronted by the Zandmotor. Your morphological data tells a similar story. Volumetric changes of foredunes at the Zandmotor are quite comparable (and even lower) than volumetric changes of foredunes in another areas (e.g., to the W). This would suggest to me that the extra accommodation space provided by the Zandmotor is not having a clear effect (at least at this temporal scale) on sediment input to coastal dunes, and that other variables (e.g., supply limiting conditions as suggested by de Vries and Honhout) are playing an important role at this location.

3. Coastal morphology section. You mention in P4-L23 that 'Changes in coastal morphology (are) expressed by average change in height per year ... ' Your figures seem to indicate that you've been calculating DTMs of difference? (i.e., simply extracting one DTM from the previous one to see spatial changes in elevation). This sounds good to me but if this is the case, then I would engage a bit more with the literature on DoDs, including the application of geomorphic change detection software to account for uncertainty in DEMs (e.g., Wheaton et al. 2010. Accounting for uncertainty in DEMs from repeat topographic surveys: improved sediment budgets. ESLP).

4. Following on the previous comment, your DTMs are fantastic in terms of their spatial extent, but 2m pixel resolutions are still quite coarse to detect some dune morphological changes in just 4 years (specially if it is dune accretion). I think it would be worth exploring the error associated to using 2 m pixel resolution DTM's in your volumetric analyses.

СЗ

Also, could you mention in the methods how are the volumes reported in Figure 4C and in the results calculated? What do you mean by 'coast'? ('beach'+'dunes'?). Are you including embryo dunes in the beach budget or in the dunes budget (or in none)?

5. Figure 4 (and associated text) – how do you morphologically define embryo dunes in this figure? The beach is defined as areas from 0-6 m and foredune from 6-12 m (if I'm getting things correctly). However, how do you characterize embryo-dunes? They are classified as '<6m' which is basically the beach too, and a height below your foredune toe? Would it be possible for you to include the procedure by which you've morphologically identified embryo dunes using DTMs (and how these are different from bedforms)? Do you think the 2m pixel resolution DTM is limiting the analyses in here?

6. Figure 5 – how did you measure dune height? This seems like a straightforward question, but could you include some details of this procedure in the methods? Did you use transects along the coast and picked the highest point at the dune crest? Or maybe contour lines, etc.?

7. There are methodological details throughout the results that should be placed within the methods section: e.g., P6, information in this page about the extent of the spatial analyses (e.g., L24) or the area covered by the domain (L19). Also, could you elaborate on the spatial extent? What is the landward limit of the area you are looking at? Is it the road or the path that seems to run landwards from the main foredune in Figure 1?

8. P4, L28 - There seems to be other type of vegetation species in Figure 1. How did you differentiate dune cover by marram grass from dune cover by other plants? (was this important?). Also, and although this is minor, references in page 5, L5 (NDVI) are good but a bit dated. Any examples of recent uses of NDVI on coastal dune vegetation? Also, could you explain how do you differentiate between vegetation cover over embryo dunes vs. foredunes? (P7, L10, or figure 4). Do you first identify embryo dunes morphologically (see comment 5) and then just simply map the vegetation data over them to separate embryo dune vegetation?

9. P7, L31 – You suggest that it is possible to 'verify' that there is a 'steady accumulation of wind-blown sand' by comparing changes in vegetation (from 2016-2017) with average yearly changes in dune heights (from 2013-2017). I am not sure I am looking at this from the correct angle, but I would argue that this assertion probably needs some field testing, or some validation using actual observations of sand transport at different locations along the coast. I wonder if there is already published material at this site giving you an order of magnitude of how much sand is moved by wind at different locations? (perhaps the work by de Vries and Honhout which you cite?). Also, vegetation can change (marram can grow or die) without morphological changes to the dunes (specially in such a short period of time of 1 year). This takes me back to L7 in this page (text corresponding to map 4D), where you propose that changes in marram grass cover can be used as a proxy for dune development (I would disagree with this because the marram can expand over 1 year without dune growth).

10. Really nice discussion on human impact and the role played by 'dynamic restoration'. I think your figure 4 and additional figure 8 are actually really good for this section, because they provide a very clear holistic view of the entire coastline, and help contextualize human impact. I think this part of the paper is important and I would leave it as is.

**Some additional comments are:**

Figure 3 – as you rightly identify in the text, the validation results for the Sentinel 2 cover are relatively weak. I agree with your explanations of figure 3, and I can see the value of using Sentinel 2 data for identifying relative trends in vegetation expansion. However, I think your results indicate that we should be rather cautions when using linear spectral unmixing over Sentinel data to derive actual values of vegetation cover. I've got two questions about this: 1) are the UAV data used in the validation vegetation density using the original 5 cm pixels or the aggregated 10 m pixels (the explanations in the current text are a bit confusing – page 5, L22 to page 6, L4); could results improve by using other type of analyses (if they exist) to derive sub-pixel proportions of

**C5**

**vegetation?**

Figure 4B – why did you choose to present elevation change data using classes with different ranges? Your first class (-1.5 to -0.25) groups pixels with elevation changes of up to -1.25 m, your next two classes cover small elevation changes of +/-0.25 m, and your fourth class covers an elevation change of +1 m. Why this particular classification? It does not allow differentiating large elevation changes (+/-1 m) from relatively lower ones.

P6, L23 - ... 'This indicates that the foredunes, since their construction in 2011, have been raised in height by 2 m due to aeolian deposition'. Could you elaborate on this? There are very large sections of the dunes in figure 4B with little to no changes (your red category). Could you indicate examples in your domain where the dunes have been raised in height?

P6, L29 – Following the first comment on Figure 4B, it is true that there seems to be a relatively continuous 'brown' line alongshore the entire domain in figure 4B. However, it is difficult to see whether this line corresponds to foredune or embryo-dunes growth. Also, it is difficult to understand the magnitude of elevation changes because 'brown' pixels are anything from 25 cm to 1.25 m. This takes me to a general comment about figure 4. I quite like it and I can see the benefits of looking at things over a large area. However, the reality is that the figure is quite difficult to interpret, and the large extent means that it is difficult to appreciate the details. I have enjoyed your arguments in terms of general trends but I think the paper would benefit from exploring different areas by sectors too? I'll leave this comment in here for you to decide.

Section 2.3.1 Linear spectral unmixing. I'm not familiar with this technique but I can see the benefits of it. However, I feel this section would benefit from including some additional references to recent work using this technique. Similar to the NDVI, these procedures are widely applied across different disciplines. Could you perhaps enrich the text by adding some recent examples of applications of these techniques to coastal

dunes or dune vegetation in general?

P9, L25 – I would recommend discussing this in the context of Davidson-Arnott et al's article (2018) on sediment budget controls on foredune height (ESPL 43).

Picky comments:

P2, paragraph starting in L7 - I quite like this paragraph and I think it is great to remind people how important coastal dune vegetation is for coastal safety. However, I wonder if you could substitute 'marram' here by 'vegetation' (specially when you talk about temperate areas in general). European marram grass is not native in many coastlines around the world where other important plants play the role of dune building.

Figure 1A –why not use a colour scheme for the 2017 elevation map? You could still overlap this over the grey area representing the extent of the 2013 Zandmotor.

P6, L19 – the area shaded in grey is not visible in Figure 4A. Perhaps there is a different way to represent this, such as a green line or other type of polyline?

In summary, a nice paper with a lot of potential. I think you've got a good amount of data and that the results are interesting, but I would work a bit more on a) trying to provide a more detailed account of the volumetric error associated with DTMs, and b) including the role played by supply-limiting conditions on the characterisation of a 'favourable' accommodation space.

Hope this helps. Do not hesitate to contact me should you need additional details, or should you want to discuss my comments. As usual, it is great to read new articles coming out of this incredible coastal experiment.

Irene Delgado-Fernandez, Edge Hill University.

---

## Author Comment (AC1) · 11 Dec 2018

We would like to thank the reviewers for providing critical feedback to our manuscript. Our apologies for not replying within the time frame set by the journal; a new job and a newborn baby (for the first author) has caused this reply to be delayed. We provided a reply to each comment which is followed by the revisions we propose to the manuscript. The text of the reviewer comments are in black, our replies are in red and proposed revisions are in blue.

M. Hilton (Referee)
University of Otago
michael.hilton@otago.ac.nz

Always fascinating to read about the efficacy of the Zandmotor and I'm sure there will be a wide audience for this paper. Specifically the authors consider the potential for this intervention to create space for the growth of marram grass and the development of embryo dunes.

They approach this question primarily from an earth science perspective. They identify the 'beach' above a height subject to storm surge and which has experienced sand accretion. Incidentally, I find it difficult to apply the term 'beach' in this context. Also, a 12-month period is a very short period to assess embryo foredune development. Was this a typical 12-month period in relation to storm surge and wind stress?

Dear Dr. Hilton, thank you for taking the time to review of our manuscript. We agree that a one year period is rather short to assess embryo / foredune development, but we would like to point out that only changes in marram grass cover has been assessed over this time period. Morphological changes have been examined over a period of three years, which may still be a relatively short period but the Zandmotor had only been constructed two years prior to that considered period. We decided to exclude the first two years of morphological changes as during those years the Zandmotor had to settle into its current behavior.

The results point to the establishment of a large accommodation space as a result of the nourishment. Foredunes have developed adjacent to the Zandmotor, presumably (and forgive me if the authors explain this) as a result of bioengineering. The development of embryonic dune across the beach, since culmination of the Zandmotor, has been patchy. Even the embryonic dunes at the back of the beach, adjacent to Duinfietspad (the road boarding Zandmotor) – some of which have been established from deliberate plantings of marram - have barely flourished. There is an alternative – ecological - perspective on this situation. A considerable section of the paper is committed to explaining how marram grass traps sand and grows by the vertical and horizontal extension of rhizome. But what the germination and growth requirements of this species? Does the surface of the Zandmotor, or most of the surface, satisfy these requirements? Further, marram growth and embryo dune development results from the deposition and germination of marram grass seed and / or rhizome. The process of marram rhizome dispersal and growth is well documented in these papers:

Konlechner, T. M., Hilton, M. J., & Orlovich, D. A. (2013). Accommodation space limits plant invasion: Ammophila arenaria survival on New Zealand beaches. Journal of Coastal Conservation, 17(3), 463-472. doi: 10.1007/s11852-013-0244-5.

Konlechner, T. M., Orlovich, D. A., & Hilton, M. J. (2016). Restrictions in the sprouting ability of an invasive coastal plant, Ammophila arenaria, from fragmented rhizomes. Plant Ecology, 217(5), 521-532. doi: 10.1007/s11258-016-0597-6.

In short, if marram propagules are not being deposited on the surface of the Zandmotor we cannot expect the development of dunes. Moreover, a range of conditions will limit the germination of seed and the success of growth from rhizome. In the case of the former the seed must remain close to the surface (within 7cm) to avoid dormancy (to germinate and to avoid entering the long-term seedbank). A very active substrate, where there are high rates of sedimentation, may also result in unfavourable growth conditions, even if germination occurs. A wind rose was not included in the paper, but I recall the drift potential is generally to the northeast . . . so there is no source of seed upwind of the Zandmotor? Similarly, the likelihood of the waves depositing rhizome on the beach is low or nil because of the elevation of the beach.

Marram growth from either seed or rhizome is inhibited by high moisture content and the surface of the beach appears to be corrugated with alternating transverse protodunes and moist deflation areas. One last (ecological) point. Marram requires nutrients to grow and so growth is usually most vigorous immediately adjacent to a beach, where decomposing algae and marine debris provide abundant fuel for growth. I wonder what the nutrient status of the substrate of the Zandmotor might be? It is probably quite low. Therefore, if I'm correct, the actual accommodation space generated by the Zandmotor might be a good deal less than you estimate. I would like to see some experimental work on the Zandmotor, with the goal of examining the biological requirements of marram grass and the dispersal of propagules. Maybe the authorities should plant more marram? But where to maximise success? It may not be in the areas experiencing the most rapid accretion! Because in these areas the rate of accretion may exceed the tolerance of seedlings and be too dry for growth from rhizome (which would not be deposited at these sites in any case). A couple of other issues. I found the maps very difficult to interpret (although I was dealing with the grey-scale versions). I very much enjoyed reading about the coast to the north and the constraints imposed by use of the coast. Finally, my apologies for the time it has taken me to comment. I value the paper and it is a worth contribution provided you acknowledge you are looking at a discrete element.

We agree that the conditions required for successful establishment of marram grass deserves more attention in our manuscript. We have therefore made changes throughout the manuscript to emphasize that dune development potential along the Delfland coast may not be fully realized in part due to conditions that may hamper marram grass establishment and thus dune development. Most prominently, we propose to include the following paragraph in the discussion:

Marine forcing, at the same time, has been shown to be an important agent in the dispersal of marram grass rhizome fragments and subsequent dune establishment via clonal growth \citep{konlechner2009potential,hilton2011incipient}. The distribution of (embryo) dunes on the southern part of the Zandmotor, as shown in close-up in fig.~\ref{fig_08}, suggests a correlation to marine dispersal of rhizome fragments as a large number of embryo dunes are present between the identified boundary height of 1.6~m~+MSL and the (current) maximum expected storm surge height of 4~m~+MSL. The embryo dunes around the small dune lake, while technically located in the same elevation zone, have likely established mostly by seed germination as the high constructed base of the \textit{Zandmotor} has completely blocked all storm surge impact until now. Over the years there has been a steady build-up of a freshwater lens under the Zandmotor and the salinity of the dune lake has significantly decreased as a result \citep{huizer2016fresh}. This fresh water availability, in combination with moderate burial dynamics, have been shown by \citet{konlechner2013accommodation} to be beneficial to marram grass seed germination and subsequent dune establishment. The specific distribution of embryo dunes around the dune lake may therefore correlate best to seed dispersal by wind coming from the dominant south-western wind direction, either pushing the seeds over the lake towards the north-east corner of the lake or depositing it on the south-west lee side where the beach slopes downwards towards the lake. In effect, fig.~\ref{fig_08} illustrates that, even though the Zandmotor may provide wide favorable accommodation space and thus a high potential for dune development, the conditions required for successful (natural) dune establishment must also be considered. Having said that, \citet{puijenbroek2017dunes} showed in a field transplant experiment that planted marram grass (consisting of a rhizome fragment with one shoot) thrived on most parts of the Zandmotor except when exposed to direct wave action. This suggests that conditions that limit marram grass growth and subsequent dune development (e.g. high salinity, drought, low nutrient status) are mostly absent on the Zandmotor and likely along the entire Delfland coast.

[Figure]

*Illustration 1: The distribution of (embryo) dunes on the Zandmotor, suggesting a correlation to marine dispersal of rhizome fragments as a large number of embryo dunes are present between the identified boundary height of 1.6~m~+MSL and the (current) maximum expected storm surge height of 4~m~+MSL.}*

Delgado-Fernandez (Referee)
delgadoi@edgehill.ac.uk

Thank you for allowing me to comment on your article and apologies for the delay in sending this review. I hope my feedback below is useful and I will be happy to further discuss changes to the paper should you wish to do so. I have very much enjoyed your article, and I believe there is good material in it. I think the text needs work though and I would concentrate in two main areas: (1) the methods and (2) the definition of 'favorable' accommodation space. You have a really good dataset but the pixel resolution of both your LiDAR and Sentinel data has limitations. The error introduced by these limitations could be quantified, as this has implications for both your results/conclusions and for the maps you produce. Additionally, some of the generalizations in this article may need field validation (in my opinion), especially those related to the concept of 'favorable accommodation space'. My concern with this concept (as it is currently defined in the text) is that it does not consider surface conditions having an effect on aeolian transport along this section of the coastline (which could in fact determine whether the accommodation space is indeed 'favorable' or not) or conditions for the colonization of new marram grass seeds on the beach. It might be a matter of further discussing this in the article? (I have included some ideas below which you may want to consider). Finally, and although I can see the benefit of looking at long sections of the coast, I feel that the article has the potential of providing much more detail on the morphological evolution of different areas by, for example, zooming into particular sections of the dunes.

Dear Dr. Delgado-Fernandez, thank you for taking the time to review our manuscript. To briefly remark here already (and on which we elaborate later), we aimed to make a distinction in the manuscript between accommodation space favorable for dune development and conditions required for marram grass establishment. The identification of favorable accommodation space and relating that to dune development potential is presented in the manuscript as the main result and is mostly based on examining the physical dynamics of sand budgets and marram grass dune cover changes. In the discussion we would like to elaborate on processes and conditions that may limit the establishment of dune vegetation within the identified accommodation space even though it is considered favorable for dune development.

Here are a few major comments:

1. Very nice introduction, and great to read the contextual and historical information about this part of the Dutch coast in section 2.1. I think, however, that there is a need to include some additional information on the peculiarities of aeolian transport at the Zandmotor. The paper gravitates around 2 conditions for the development of coastal dunes: 1) steady supply of wind-blown sand; and 2) wide accommodation space (or simply beach width).

We would like to remark that beach width and accommodation space are related, but that accommodation space in our opinion is more narrowly defined as 'the space available for potential sediment accumulation'. Not all parts of a wide beach may be defined as accommodation space as it may be erosive or act as an through-put of sediment.

The second is important as it is related with fetch distances, etc. (I'm suggesting the paper by Delgado-Fernandez, 2010, on the fetch effect because it is a review article but please choose any other that may suit you well), additional to reducing storm impact as you describe in the article. However, accommodation space is only one of the many variables involved in sediment input to coastal dunes. Aeolian transport is a function of many surface conditions including the formation of lag deposits or the composition of sediment, which seems to be an important limiting aspect at the Zandmotor (as described by the group of de Vries and Honhout, etc.). This is, the Zandmotor does seem to have created accommodation space but the question is how 'favourable' this accommodation space really is for maximizing aeolian transport and hence dune growth (not necessarily just limiting dune erosion). I would recommend that potential supply-limiting conditions affecting transport by wind at this location are reviewed in the introduction as these affect your condition n. 1 ('steady supply of windblown sand').

We would like to argue that the objective of the manuscript is not to investigate 'sediment input to coastal dunes', but rather to investigate the potential of dune development along the Delfland coast according to available accommodation space and how the Sand Motor contributes to creating and maintaining this accommodation space. Favorable accommodation space for dune development, in our opinion, is not necessarily defined by areas with maximum aeolian transport but rather by areas that are have been constantly accretive over time. In that context we feel that supply-limiting conditions and the effect of fetch distance on aeolian fluxes are relevant issues but may be better suited in the context of the discussion. In the results we show that, regardless of lag deposits or potential fetch effects, large areas along the Delfland coast do receive a 'steady supply of wind-blown sand' and may thus be characterized by having accommodation space favorable for dune development.

2. The inclusion of supply limiting conditions above might have implications for your discussion. Your section 4.1 suggests that you are using the presence of vegetation as a proxy for sediment input to the dunes (something I come back to in comment n. 9). I do not think 1 year is long enough for the analyses you make. However, even if it was, your figure 4 seems to suggest that vegetation in areas other than the Zandmotor has followed a similar trend than those fronted by the Zandmotor. Your morphological data tells a similar story. Volumetric changes of foredunes at the Zandmotor are quite comparable (and even lower) than volumetric changes of foredunes in another areas (e.g., to the W). This would suggest to me that the extra accommodation space provided by the Zandmotor is not having a clear effect (at least at this temporal scale) on sediment input to coastal dunes, and that other variables (e.g., supply limiting conditions as suggested by de Vries and Honhout) are playing an important role at this location.

At P7, L7 We mention 'Using changes in marram grass cover as a proxy for dune development', which in our opinion is not the same as 'sediment input to the dunes'. Perhaps we gave that impression, but it has not been our intention to relate accommodation space provided by the Zandmotor to sediment input to coastal dunes. Our intention was to identify accommodation space along the Delfland coast favorable to dune development (potential), according to two criteria (i.e. sheltered from storm impact and receiving a steady supply of wind-blown sand). The notion that not all available accommodation space is being 'occupied' by developing dunes, may relate to conditions that constrain the establishment of dunes. We propose to elaborate on that issue in the discussion.

3. Coastal morphology section. You mention in P4-L23 that 'Changes in coastal morphology (are) expressed by average change in height per year . . . ' Your figures seem to indicate that you've been calculating DTMs of difference? (i.e., simply extracting one DTM from the previous one to see spatial changes in elevation). This sounds good to me but if this is the case, then I would engage a bit more with the literature on DoDs, including the application of geomorphic change detection software to account for uncertainty in DEMs (e.g., Wheaton et al. 2010. Accounting for uncertainty in DEMs from repeat topographic surveys: improved sediment budgets. ESLP).

The DTM's used in this study were derived from airborne lidar provided by a professional contractor operating under standardized procedures and according to strict quality guidelines. While the mentioned paper by Wheaton et al. 2010 provides an interesting read, we feel that the outlined procedure (to account for uncertainty in DEMs from repeat topographic surveys) applies more to DEM's that were derived from manual topographic surveys rather than DEM's that were derived by professional airborne lidar. The focus on manual error is also pointed out in the paper at P139 for instance: "In areas where no agent for geomorphic change was experienced (i.e. high island and terrace surfaces not subject to inundation, flooding or significant overland flow), evidence for change is clearly questionable. However, small elevation differences are not surprising and reflect a combination of random surveying errors (e.g. GPS triangulation errors, pole tilt, etc.), systematic errors (e.g. resection errors, interpolation errors, incorrect survey rod height, etc.), the limits of the instrument precision, operator blunders, and sampling differences between surveys."

4. Following on the previous comment, your DTMs are fantastic in terms of their spatial extent, but 2m pixel resolutions are still quite coarse to detect some dune morphological changes in just 4 years (specially if it is dune accretion). I think it would be worth exploring the error associated to using 2 m pixel resolution DTM's in your volumetric analyses.

We agree that this may provide a valuable contribution. We were, however, not able to verify the vertical error associated with the DTM's due to lack of absolute elevation data: the contractor responsible for the Lidar flights in fact also provides the official elavation data for the entire Netherlands. We did conduct a quality check on the comparability of the different DTM's. We propose to briefly mention this in the section 2.2. coastal morphology:

The contractor responsible for the lidar flights guaranteed a minimum density resolution of 1 laser point per square meter, with a systemic vertical error equal or less than 2.6 cm and standard deviation equal or less than +/- 2.0 cm. The quality of the DTM's were verified by comparing the height of features along the coastline that remained unchanged between 2013 and 2017 (e.g. parking lots, paved roads, building rooftops). The average standard deviation of the vertical component of the five DTM's (using approx. 100 samples) was determined at +/- 2.4 cm.

Also, could you mention in the methods how are the volumes reported in Figure 4C and in the results calculated?

Volumetric changes were calculated by the surface area of each pixel (4 m^2) multiplied by the yearly change in height of that pixel. Changes in sand volume are either reported as a (total) volume per year (m^3/y) or as an alongshore volume (m^3 /m/y), as volume per meter coastline per year. We propose to better clarify this in the methods section as following:

Changes in coastal morphology were first expressed by average change in height per year (m/y), which was calculated per consecutive time-step: t51 =(t21 + t32 + t43 + t54) / 4. This was done to better consider temporal variations within each year and to account for yearly changes of the shoreline. Coastal morphological changes were also expressed as an average (alongshore) change in sand volume per year (m^3/y or m^3/m/y). Sand volume changes were likewise calculated per consecutive time-step and obtained by multiplying the surface area of each pixel (4 m^2) by its yearly change in height.

What do you mean by 'coast'? ('beach'+'dunes'?). Are you including embryo dunes in the beach budget or in the dunes budget (or in none)?

What we intended to show in Fig 4C is the sand budget for the whole of the Delfland coast ('coast') and differentiated by the beach and foredunes (where beach + foredunes = coast). The embryo dunes are included in the sand budget for the beach because their volumetric changes were too small to depict at the scale used in the figure and are features present on the beach. To more clearly describe Fig 4C I propose to change the caption below the figure to:

Average alongshore changes in coastal sand volumes (m^3/m/y) between 2013 and 2017 for the Delfland coast and differentiated by the beach (embryo dunes included) and foredunes.

5. Figure 4 (and associated text) – how do you morphologically define embryo dunes in this figure? The beach is defined as areas from 0-6 m and foredune from 6-12 m (if I'm getting things correctly). However, how do you characterize embryo-dunes? They are classified as '<6m' which is basically the beach too, and a height below your foredune toe? Would it be possible for you to include the procedure by which you've morphologically identified embryo dunes using DTMs (and how these are different from bedforms)? Do you think the 2m pixel resolution DTM is limiting the analyses in here?

The embryo dunes on the beach were identified using the (2017) Sentinel-2 imagery rather than the lidar-derived DTM's. Because the beach and the foredunes were completely reconstructed between 2009 and 2011 it was relatively straightforward to distinguish between the foredunes that were planted by marram grass and embryo dunes that developed naturally on the beach after the construction was finished.

6. Figure 5 – how did you measure dune height? This seems like a straightforward question, but could you include some details of this procedure in the methods? Did you use transects along the coast and picked the highest point at the dune crest? Or maybe contour lines, etc.?

All data on morphology and morphological changes of the Delfland coast were obtained from the DTM's. The presence of marram grass within a 10-meter Sentinel pixel was used to indicate the presence of dunes (both foredunes and embryo dunes).

7. There are methodological details throughout the results that should be placed within the methods section: e.g., P6, information in this page about the extent of the spatial analyses (e.g., L24) or the area covered by the domain (L19). Also, could you elaborate on the spatial extent? What is the landward limit of the area you are looking at? Is it the road or the path that seems to run landwards from the main foredune in Figure 1?

The landward limit of the area has indeed not been properly defined. We propose the following clarification in the manuscript:

P4, L32-34: The Delfland coast considered in the analysis covers an subaerial area of about 500 ha and extends in the landward direction until the older established dunes. This approximately coincides with the paved bike-path (gray line in fig.~\ref{fig_01}) running along the crest of the newly created foredune ridge.

8. P4, L28 - There seems to be other type of vegetation species in Figure 1. How did you differentiate dune cover by marram grass from dune cover by other plants? (was this important?). Also, and although this is minor, references in page 5, L5 (NDVI) are good but a bit dated. Any examples of recent uses of NDVI on coastal dune vegetation? Also, could you explain how do you differentiate between vegetation cover over embryo dunes vs. foredunes? (P7, L10, or figure 4). Do you first identify embryo dunes morphologically (see comment 5) and then just simply map the vegetation data over them to separate embryo dune vegetation?

The foredunes that were included in the study were covered solely by marram grass. We were able to ascertain this from the aerial photos and Sentinel-2 imagery. We already mention at P6, L24-25

that ' Older established dunes were excluded from the analysis as they are minimally exposed to marine forces and mostly covered with vegetation species other than marram grass.' We propose moving this section to the methodology for dune vegetation cover, and mention that this was clear by using a cutoff value for NDVI of 0.6:

Older established dunes (with NDVI~>~0.6 in Fig.~\ref{fig_02}) were excluded from the analysis as they are minimally exposed to marine forces and mostly covered with vegetation species other than marram grass. Further, all man-made structures on the beach related to coastal safety (e.g. groynes) and leisure and recreation were masked before the linear spectral unmixing procedure was executed.

9. P7, L31 – You suggest that it is possible to 'verify' that there is a 'steady accumulation of wind-blown sand' by comparing changes in vegetation (from 2016-2017) with average yearly changes in dune heights (from 2013-2017). I am not sure I am looking at this from the correct angle, but I would argue that this assertion probably needs some field testing, or some validation using actual observations of sand transport at different locations along the coast. I wonder if there is already published material at this site giving you an order of magnitude of how much sand is moved by wind at different locations? (perhaps the work by de Vries and Honhout which you cite?). Also, vegetation can change (marram can grow or die) without morphological changes to the dunes (specially in such a short period of time of 1 year). This takes me back to L7 in this page (text corresponding to map 4D), where you propose that changes in marram grass cover can be used as a proxy for dune development (I would disagree with this because the marram can expand over 1 year without dune growth).

At P7, L25-26 we re-iterate that we consider accommodation space favorable for dune development when when it is (1) sheltered from storm impact and (2) experiencing a steady accumulation of wind-blown sand. What we meant by 'verifying' the second condition is to check whether dunes covered by marram grass were indeed predominantly present in accreting areas and whether these dunes increased in cover between marram grass. So the aim was not to verify a 'steady accumulation of wind-blown sand' but whether favorable accommodation space can be characterized by having that condition. Comparing the changes in dune cover by marram grass from 2016 to 2017 to the average yearly change in dune height between 2013 – 2017 seems to indicate that this is indeed the case. We propose to clarify the meaning of 'verify' as follows:

The second condition is identified (or verified as the positive effect of sand burial on marram grass vigor is well documented) by comparing the changes in dune cover by marram grass between 2016 - 2017 to the average yearly change in dune height between 2013 – 2017

We agree that changes in marram grass cover do not necessarily translate to changes in dune development. We therefore propose to insert the word 'potential'

10. Really nice discussion on human impact and the role played by 'dynamic restoration'. I think your figure 4 and additional figure 8 are actually really good for this section, because they provide a very clear holistic view of the entire coastline, and help contextualize human impact. I think this part of the paper is important and I would leave it as is.

Some additional comments are:
Figure 3 – as you rightly identify in the text, the validation results for the Sentinel 2 cover are relatively weak. I agree with your explanations of figure 3, and I can see the value of using Sentinel 2 data for identifying relative trends in vegetation expansion.

However, I think your results indicate that we should be rather cautions when using linear spectral unmixing over Sentinel data to derive actual values of vegetation cover. I've got two questions about this: 1) are the UAV data used in the validation vegetation density using the original 5 cm pixels or the aggregated 10 m pixels (the explanations in the current text are a bit confusing – page 5, L22 to page 6, L4); could results improve by using other type of analyses (if they exist) to derive sub-pixel proportions of vegetation?

Dune cover was calculated for the resampled 10 m pixels but expressed as the proportion of 5 cm pixels classified as marram grass within each aggregated 10 m pixel. To clarify this in the manuscript we propose the following paragraph:

Using a K-means clustering algorithm \citep{hartigan1979algorithm}, the individual 5 cm pixels of the orthomosaic were classified either as beach sand or marram grass. The accuracy of the algorithm was confirmed by visual inspection; for more details about acquisition and processing of the UAV-derived data the reader is referred to \citet{nolet2017uav}. The orthomosaic was subsequently resampled to match the 10 m pixel size resolution of the Sentinel-2 imagery and dune cover was calculated as the proportion of (former) 5 cm pixels classified as marram grass contained within each newly aggregated 10 m pixel.

Figure 4B – why did you choose to present elevation change data using classes with different ranges? Your first class (-1.5 to -0.25) groups pixels with elevation changes of up to -1.25 m, your next two classes cover small elevation changes of +/-0.25 m, and your fourth class covers an elevation change of +1 m. Why this particular classification? It does not allow differentiating large elevation changes (+/-1 m) from relatively lower ones.

After closer inspection we realized that the lowest (-1.5m) and highest (1.25m) value in the map are in the 1$^{st}$ and 99$^{th}$ quantile of the distribution of average height differences and are thus most likely outliers. We therefore limited the range of the classes based on the values falling within the 1-99 quantile (in a similar fashion as is done in GIS software such as QGIS). Now the lowest class starts at -0.75m and the highest ends at 0.5m, making the range of the classes more balanced overall.

P6, L23 - . . . 'This indicates that the foredunes, since their construction in 2011, have been raised in height by 2 m due to aeolian deposition'. Could you elaborate on this? There are very large sections of the dunes in figure 4B with little to no changes (your red category). Could you indicate examples in your domain where the dunes have been raised in height?

In the introduction we mention (P4, L1-2) that "After completion the newly created dune ridge ranged between 4-12 meter above mean sea level (m+MSL) from toe to crest." This information is taken from literature but also checked using a DTM from 2011. What we fail to clarify is that this suggests that the 'the toe of the foredunes, compared to their construction height in 2011, have been raised by about 2 m due to aeolian deposition'. We propose to insert this sentence into the manuscript. The dark red category in Fig 4B ranges between an average height change of 0.25-0.5 m/y (changed after previous point), so in the course of 5 years a height change between 1.25 and 2.5 can be expected.

P6, L29 – Following the first comment on Figure 4B, it is true that there seems to be a relatively continuous 'brown' line alongshore the entire domain in figure 4B. However, it is difficult to see whether this line corresponds to foredune or embryo-dunes growth. Also, it is difficult to understand the magnitude of elevation changes because 'brown'

pixels are anything from 25 cm to 1.25 m. This takes me to a general comment about figure 4. I quite like it and I can see the benefits of looking at things over a large area. However, the reality is that the figure is quite difficult to interpret, and the large extent means that it is difficult to appreciate the details. I have enjoyed your arguments in terms of general trends but I think the paper would benefit from exploring different areas by sectors too? I'll leave this comment in here for you to decide.

We have included a figure (fig. 8) showing the Zandmotor in more detail in order to relate changes in morphology and dune cover to conditions required (and responsible) for establishment of marram grass.

Section 2.3.1 Linear spectral unmixing. I'm not familiar with this technique but I can see the benefits of it. However, I feel this section would benefit from including some additional references to recent work using this technique. Similar to the NDVI, these procedures are widely applied across different disciplines. Could you perhaps enrich the text by adding some recent examples of applications of these techniques to coastal dunes or dune vegetation in general?

We have added the following citations to the manuscript:

Linear spectral unmixing
- Lucas, N. S., Shanmugam, S., & Barnsley, M. (2002). Sub-pixel habitat mapping of a costal dune ecosystem. Applied Geography, 22(3), 253-270.
- Zhang, L., & Baas, A. C. (2012). Mapping functional vegetation abundance in a coastal dune environment using a combination of LSMA and MLC: a case study at Kenfig NNR, Wales. International journal of remote sensing, 33(16), 5043-5071.

NDVI
- Pettorelli, N., Vik, J. O., Mysterud, A., Gaillard, J. M., Tucker, C. J., & Stenseth, N. C. (2005). Using the satellite-derived NDVI to assess ecological responses to environmental change. Trends in ecology & evolution, 20(9), 503-510.

P9, L25 – I would recommend discussing this in the context of Davidson-Arnott et al's article (2018) on sediment budget controls on foredune height (ESPL 43).

We have included the reference in the discussion.

Picky comments:

P2, paragraph starting in L7 – I quite like this paragraph and I think it is great to remind people how important coastal dune vegetation is for coastal safety. However, I wonder if you could substitute 'marram' here by 'vegetation' (specially when you talk about temperate areas in general). European marram grass is not native in many coastlines around the world where other important plants play the role of dune building.

We would prefer to keep the focus on marram grass as this is the coastal engineer responsible for dune development in our study area.

Figure 1A –why not use a colour scheme for the 2017 elevation map? You could still overlap this over the grey area representing the extent of the 2013 Zandmotor.

P6, L19 – the area shaded in grey is not visible in Figure 4A. Perhaps there is a different way to represent this, such as a green line or other type of polyline?

In summary, a nice paper with a lot of potential. I think you've got a good amount of data and that the results are interesting, but I would work a bit more on a) trying to provide a more detailed account of the volumetric error associated with DTMs, and b) including the role played by supply-limiting conditions on the characterisation of a 'favourable' accommodation space.

Hope this helps. Do not hesitate to contact me should you need additional details, or should you want to discuss my comments. As usual, it is great to read new articles coming out of this incredible coastal experiment.
Irene Delgado-Fernandez, Edge Hill University.

[revised manuscript text omitted]

**2.2.1  Linear spectral unmixing**

The four selected bands were stacked into a new multispectral data cube and a linear spectral unmixing procedure was ap-
plied. This was done to derive sub-pixel proportions of dune cover by marram grass, the most prominent and abundant dune-building species. Linear spectral unmixing is an approach to determine the relative abundance of user-specified ground cover components (endmembers) in multispectral (or hyperspectral) imagery based on its spectral characteristics (e.g., Smith et al., 1985; Settle and Drake, 1993; Theseira et al., 2003). It has successfully been applied before by Lucas et al. (2002) and Zhang and Baas (2012) in mapping the abundance of vegetation, including marram grass, in coastal dune environments. The reflectance at each pixel of the image is assumed to be a linear combination of the reflectance of the endmembers present within the pixel:

$$R_i = \sum_{k=1}^{n} (f_k \, R_{ik}) + e_i \qquad (1)$$

Where $i = 1, ..., m$ are the number of spectral bands, $R_i$ is the reflectance of band $i$ of each pixel, $k = 1, ..., n$ are the number of endmembers, $f_k$ is the proportion of endmember $k$ within each pixel, $R_{ik}$ the spectral reflectance of endmember $k$ within each pixel on band $i$, and $e_i$ is the residual error term (Lu et al., 2003). Here, two endmembers were specified (see Fig. 2). The first endmember was made up by a group of pixels ($\sim 8$) containing only beach sand, the second endmember by a similarly sized group of pixels fully covered by marram grass. The spectra of the two endmembers were obtained for each Sentinel-2 image separately, and maps containing sub-pixel proportions of beach sand and marram grass were derived using ENVI version

4.8 (Exelis Visual Information Solutions, Boulder, Colorado). The sub-pixel proportions of marram grass were subsequently interpreted as a percentage dune cover within each 10 meter pixel. Older established dunes (with NDVI > 0.6 in Fig. 2) were excluded from the analysis as they are minimally exposed to marine forces and mostly covered with vegetation species other than marram grass. Further, all man-made structures on the beach related to coastal safety (e.g. groynes) and leisure and recreation were masked from the imagery before the linear spectral unmixing procedure was executed.

Changes in dune cover by marram grass along the Delfland coast were obtained by subtracting the percentages dune cover calculated for the 2016 Sentinel-2 image from the snapshot of 2017. Changes in dune cover between 2016 and 2017 were expressed for every 10-meter pixel but also as an alongshore change in cover area (m²/m/y). This was done for better interpretation of dune dynamics along the Delfland coast and was calculated by multiplying the surface area of each pixel (100 m²) by its fractional cover change. The linear spectral unmixing procedure was validated against a high-resolution orthophoto orthomosaic of a stretch of foredune directly adjacent to the *Zandmotor* (see also fig. 3). The georeferenced orthophoto orthomosaic (5 cm pixel size) was obtained by an Unmanned Aerial Vehicle (UAV) during a flight on 01-09-2016 September 1 2016, so 10 days before the acquisition date of the 2016 Sentinel-2 image. Using a *k*-means clustering algorithm (Hartigan and Wong, 1979), individual pixels the individual 5 cm pixels of the orthomosaic were classified either as beach sand or marram grass. The accuracy of the algorithm was confirmed by visual inspection. The 5 cm pixels were subsequently aggregated into individual ; for more details about acquisition and processing of the UAV-derived data the reader is referred to Nolet et al. (2017). The orthomosaic was subsequently resampled to match the 10 m pixels pixel size resolution of the Sentinel-2 imagery and dune cover depicted in the orthomosaic was calculated as the proportion of the (former) 5 cm pixels classified as marram grass within each contained within each newly aggregated 10 m pixel.

**FIGURE 2 ABOUT HERE.**

**2.3    Coastal morphology**

Data on the morphology and morphological changes of the Delfland coast were obtained from Digital Terrain Models (DTM) provided by Rijkswaterstaat, the executive agency of the Ministry of Infrastructure and Water Management. The 2 m pixel size

DTM's are produced every year (since 1996) for coastline monitoring purposes by airborne Lidar and have been made public under a Creative Commons Zero (CC0) statement. The contractor responsible for the lidar flights guaranteed a minimum density resolution of 1 laser point per square meter, with a systemic vertical error equal or less than 2.6 cm and standard deviation equal or less than $\pm$ 2.0 cm. Five yearly DTM's of the Delfland coast were used for analysis, acquired in spring 2013 till spring 2017. Changes in coastal morphology were first expressed by average change in height per year (m/y), which was calculated per consecutive time-step: $t_{5-1} = (t_{2-1} + t_{3-2} + t_{4-3} + t_{5-4})/4$. This was done to better consider temporal variations within each year and to account for yearly changes of the shoreline. Coastal morphological changes were also expressed as an average (alongshore) change in sand volume per year (m$^3$/y or m$^3$/m/y). Sand volume changes were likewise calculated per consecutive time-step and obtained by multiplying the surface area of each pixel (4 m$^2$) by its yearly change in height. The quality of the DTM's were verified by comparing the height of features along the coastline that remained unchanged between 2013 and 2017 (e.g. parking lots, paved roads, building rooftops). The average standard deviation of the vertical component of the five DTM's (using $\approx$ 100 samples) was determined at $\pm$ 2.4 cm. Data are available at: https://rijkswaterstaat.nl/apps/geoservices/geodata/dmc. The Delfland coast considered in the analysis covers an subaerial area of about 500 ha and extends in the landward direction until the older established dunes. This approximately coincides with the paved bike-path (gray line in fig. 1) running along the crest of the newly created foredune ridge. In order to compare the morphology and morphological changes of the Delfland coast against the presence of dunes and changes in dune cover by marram grass, the 2 m resolution DTM's were resampled using bilinear interpolation to match the 10 m pixel size of the Sentinel-2 imagery.

**3   Results**

Figure 3 shows the results of validating the linear spectral unmixing procedure on the Sentinel-2 images. The dune cover calculated from the orthophoto and the two Sentinel-2 images are plotted against each other in fig. 3B. It is clear that deriving sub-pixel proportions of dune cover using linear spectral unmixing result in an overestimation of dune cover by marram grass. Even though 54% of the variance for 2016 Sentinel-2 image can be explained by a positive linear regression model, most of the data points deviate from the 1:1 identity line because of higher dune cover values calculated by the Sentinel-2 image. This trend, however, appears to be systematic to the linear spectral unmixing procedure since the data points from the 2017 Sentinel-2 image deviate even further from the identity line. This lower correlation ($R^2 \approx 0.35$) is in line with expectation as dune cover by marram grass was observed to have increased at this location between 2016 and 2017. So even though the linear spectral unmixing procedure overestimates the sub-pixel proportions of dune cover, the derived marram grass cover values for each Sentinel-2 image appear to be comparable relative to each other.

**FIGURE 3 ABOUT HERE.**

Figure 4 shows the derived maps used to identify favorable accommodation space for dune development.The

 first map (4A) gives an overview of the morphological features, including the *Zandmotor*, during early spring 2017. The beach ranges between 0 - 6 m +MSL in height and this is where new embryo dunes have either formed or expanded since 2011. The foredunes are exclusively covered by marram grass and stretch fully along the coastline, albeit at variable widths,  at heights between 6 - 14 m +MSL. This indicates that the  toe of the foredunes, compared to their construction height in 2011, have been raised  by about 2  m due to aeolian deposition.  The second map (4B) shows how the subaerial coastal morphology changed between 2013 and 2017, expressed by the average yearly change in height (m/y). It is clear that the most seaward part of the *Zandmotor* experienced strong erosion due to marine forcing, while most of the foredunes and particularly the beach just south and north of the *Zandmotor* experienced accretion due sand spreading effects. The base of the *Zandmotor* with its high construction height either remained relatively stable or experienced moderate erosion. 
[revised manuscript text omitted]
 along the Delfland coast. The results indicate that the *Zandmotor* itself provides the most favorable accommodation space, for it has large areas located above 1.6 m +MSL that  on average experience a continuous accretion by wind-blown sand. As such, the results highlight that the *Zandmotor* supports an especially high potential for new embryo dunes to develop as most of its  accommodation space is located on the beach. This section examines the merit of the identified conditions for when accommodation space is considered favorable for dune development, as well as the merit of using favorable accommodation space to indicate  dune development potential. The latter is examined in relation to the design and intended dynamical nature of the *Zandmotor*, the conditions required for successful establishment of marram grass and the persistent anthropogenic disturbances along the Delfland coast arising from recreation and  nature management practices.

**4.1  Conditions indicating favorable accommodation space for dune development**

Accommodation space is considered favorable for dune development when it is sheltered from storm impact and experiences a steady accumulation of wind-blown sand. The latter condition is not disputed, as the reinforcing feedback between the growth response of marram grass and burial by wind-blown sand is well documented (Huiskes, 1979; Disraeli, 1984; Maun and Lapierre, 1984; Van der Putten et al., 1988; Hesp, 1991; Maun, 1998) and recognized to be fundamental to coastal dune development in temperate regions around the world (e.g., Baas and Nield, 2010; Durán and Moore, 2013; Keijsers et al., 2016; Nolet et al., 2017). The positive feedback mechanism  originates from a trait that all beach grasses of the genus *Ammophila* possess, namely potentially unlimited horizontal and vertical growth through its rhizomes (Gemmell et al., 1953; Ranwell, 1972). Whether marram grass grows horizontally or vertically subsequently depends on the amount of wind-blown sand, which  makes its so particularly advantageous to dune-building. After establishment, by seed or rhizome dispersal, marram grass first produces leafy shoots along newly developing horizontal rhizomes. When wind-blown sand is trapped by the leafy shoots, the immediate sand surface is raised and a small embryo dune is formed (Hesp, 1989). The leafy shoots are capable of growing up through a moderate thickness of sand by elongation of individual leaves. If, however, a leafy shoot is overwhelmed by sand deposition, one or more of its axillary buds develop into a vertical rhizome that will continue to grow until the surface is reached. Adventitious roots are produced from the nodes of the vertical rhizome and the horizontal rhizomes gradually die, so that the vertical rhizomes become independent of one another. This process may be repeated as long as aeolian supply is abundant and marram grass continues to trap sand. The capacity to trap sand, as noted before, is enhanced by the growth response of marram grass to sand trapping, which introduces the positive feedback mechanism driving coastal dune development (Gemmell et al., 1953; Ranwell, 1972). Using very high-resolution data, Nolet et al. (2017) showed that marram grass on foredunes along the *Zandmotor*  appears to thrive best under a sand trapping rate of approximately 0.3 meter of sand per growing season and that marram grass can withstand sand burial up to 1 meter of sand. However, while this demonstrates how positive plant-sand feedback steers dune development, it must be noted that the physical size of a developing dune and predominant wind regime also controls its morphology (Davidson-Arnott et al., 2018). As dunes grow, for example, a limit is imposed on its height because the wind force required to transport sand upslope increases significantly (e.g., Arens et al., 1995; Arens, 1996; Keijsers et al., 2015a).  Coastal foredunes therefore tend to expand in width rather than height, which emphasizes the importance of the wide favorable accommodation space  the *Zandmotor* provides for foredune development.

The condition that accommodation space is considered favorable when it is sheltered from storm impact warrants closer inspection, because the impact of a storm surge depends both on the magnitude of the storm as well as the geometry of the beach (Houser et al., 2008). Wind stress due to atmospheric pressure differences drive storm surge levels and offshore wave conditions, but the vertical dimension of the beach profile, in particular, exerts great control on shoreline parameters such as wave setup, swash and runup (e.g., Stockdon et al., 2006; Sallenger Jr, 2000; Ruggiero et al., 2001). This is significant because the dissipation of kinetic energy of breaking waves is responsible for the highest rates of coastal erosion and dune decline (e.g., Vellinga, 1982; Short and Hesp, 1982). However, while empirical models can calculate wave runup levels and wave breaking energy from parameters such as offshore wave conditions and beach profile (see Stockdon et al. (2006) and Sallenger Jr (2000) for details), those relations only return approximations as often not all required model input is available or because of inherent model uncertainties. Having said that, the results suggest that dunes along the Delfland coast are sheltered from storm impact above a beach height of 1.6 m +MSL. This finding is examined in relation to offshore sea water levels measured by a buoy in close proximity to the *Zandmotor* mega-scale beach nourishment. Figure 7 shows the probability density curve (which is bimodal because of tidal dynamics) of those sea water levels (in m ± MSL), measured every 10 minutes from 2011 until 2017. Included are the instances when sea water levels exceeded the apparent 1.6 m +MSL boundary height for dunes to be sheltered from storm impact. It is clear from fig. 7 that this did not occur frequently, only during about 0.4% of the measurements. Those measurements, however, were relatively clustered together, meaning that the boundary height was exceeded over (relatively) prolonged periods of time. Although, over the course of six years this happened for no more than 10 full days. On average the exceedance was about 20 cm up to a sea water level of 1.8 m +MSL, but on a few occasions sea water levels almost doubled compared the boundary height to 3.10 m +MSL. This is excluding the wave runup onto the beach, which can be significant for natural beaches in the Netherlands. Dependent on whether the beach profile is dissipative or reflective, both Stockdon et al. (2006) and Poortinga et al. (2015) show that wave runup may reach to heights from 0.85 to 1.45 m above still water level (m), which is the level that would occur in the absence of waves. This implies that, since the construction of the *Zandmotor* in 2011, the Delfland coast may have experienced coastal erosion by storm surge levels  reaching heights up to at least 4 m +MSL.

The observation that in 2017 quite a large number of embryo dunes were present on the beach at heights well below the maximum experienced storm surge levels, points to the capacity of established dunes to withstand and recover from hydrodynamic storm impact  as well as to the pivotal role  marine dispersal of rhizome fragments likely plays to dune establishment processes. As remarked by various researchers (e.g., Suanez et al., 2012; Feagin et al., 2015; Houser et al., 2015; Van Puijenbroek et al., 2017a, b), the ability of embryo dunes to recover from storm impact largely depends to what the extent the above- and belowground structural integrity of marram grass remains intact after a storm event. This depends, in turn, on the severity of the storm impact on the dune, which can be caused by wave erosion (scarping and overwash) and swash inundation (Sallenger Jr, 2000; Hesp and Martínez, 2007). Wave erosion may completely remove all sand from an embryo dune (so it is no longer raised from the beach surface) and have an abrasive effect on the leaves of marram grass, causing either minor damage or compete removal of all aboveground biomass. Most of the belowground root system of marram grass, however, has been observed to largely remain intact after wave scarping or overwash (Feagin et al., 2015). Potential damage of swash inundation  to marram grass depends on the duration of the inundation period, but as Vergiev et al. (2013) demonstrate, marram grass displays no visible decomposition of stems, roots or rhizomes after being immersed with sea water for 20 days. This is well beyond the period a beach will be inundated after a storm event, which implies that inundation has a limited, if any, negative effect on the structural integrity of marram grass. Given that storm events occur more frequently in winter, it has been observed that embryo dunes on dissipative beaches undergo a classic seasonal cycle of erosion during the winter and accretion during the summer (Montreuil et al., 2013; Van Puijenbroek et al., 2017b). Their presence on the beach, however, would remain persistent throughout the year and often show a yearly net growth when aeolian supply was sufficient (Anthony et al., 2007; Suanez et al., 2012). This not only indicates that embryo dunes have the capacity to withstand storm impact and quickly recover to prestorm conditions, but also that  the above- and belowground structure of marram grass often remains largely intact after a storm event. Marine forcing, at the same time, has been shown to be an important agent in the dispersal of marram grass rhizome fragments and subsequent dune establishment via clonal growth (Konlechner and Hilton, 2009; Hilton and Konlechner, 2011). The distribution of (embryo) dunes on the southern part of the *Zandmotor*, as shown in close-up in fig. 8, suggests a correlation to marine dispersal of rhizome fragments as a large number of embryo dunes are present between the identified boundary height of 1.6 m +MSL and the (current) maximum expected storm surge height of 4 m +MSL. The embryo dunes around the small dune lake, while technically located in the same elevation zone, have likely established mostly by seed germination as the high constructed base of the *Zandmotor* has completely blocked all storm surge impact until now. Over the years there has been a steady build-up of a freshwater lens under the *Zandmotor* and the salinity of the dune lake has significantly decreased as a result (Huizer et al., 2016). This fresh water availability, in combination with moderate burial dynamics, have been shown by Konlechner et al. (2013) to be beneficial to marram grass seed germination and subsequent dune establishment. The specific distribution of embryo dunes around the dune lake may therefore correlate best to seed dispersal by wind coming from the dominant south-western wind direction, either pushing the seeds over the lake towards the north-east corner of the lake or depositing it on the south-west lee side where the beach slopes downwards towards the lake. In effect, fig. 8 illustrates that, even though the *Zandmotor* may provide wide favorable accommodation space and thus a high potential for dune development, the conditions required for successful (natural)
dune establishment must also be considered. Having said that, Puijenbroek (2017) showed in a field transplant experiment that planted marram grass (consisting of a rhizome fragment with one shoot) thrived on most parts of the *Zandmotor* except when exposed to direct wave action. This suggests that conditions that limit marram grass growth and subsequent dune development (e.g. high salinity, drought, low nutrient status) are mostly absent on the *Zandmotor* and likely along the entire Delfland coast.

[revised manuscript text omitted]

Davidson-Arnott, R., Hesp, P., Ollerhead, J., Walker, I., Bauer, B., Delgado-Fernandez, I., and Smyth, T.: Sediment budget controls on foredune height: Comparing simulation model results with field data, Earth Surface Processes and Landforms, 2018.

De Jong, B., Keijsers, J. G. S., Riksen, M. J. P. M., Krol, J., and Slim, P. A.: Soft Engineering vs. a Dynamic Approach in Coastal Dune Management: A Case Study on the North Sea Barrier Island of Ameland, The Netherlands, Journal of Coastal Research, 2014.

De Schipper, M. A., De Vries, S., Ruessink, G., De Zeeuw, R. C., Rutten, J., Van Gelder-Maas, C., and Stive, M. J. F.: Initial spreading of a mega feeder nourishment: Observations of the Sand Engine pilot project, Coastal Engineering, 111, 23–38, 2016.

De Vriend, H. J., van Koningsveld, M., Aarninkhof, S. G., De Vries, M. B., and Baptist, M. J.: Sustainable hydraulic engineering through
building with nature, Journal of Hydro-environment research, 9, 159–171, 2015.

Disraeli, D. J.: The effect of sand deposits on the growth and morphology of Ammophila breviligulata, The Journal of Ecology, pp. 145–154, 1984.

Drusch, M., Del Bello, U., Carlier, S., Colin, O., Fernandez, V., Gascon, F., Hoersch, B., Isola, C., Laberinti, P., Martimort, P., et al.: Sentinel-2: ESA's optical high-resolution mission for GMES operational services, Remote sensing of Environment, 120, 25–36, 2012.

Durán, O. and Moore, L. J.: Vegetation controls on the maximum size of coastal dunes, Proceedings of the National Academy of Sciences, 110, 17 217–17 222, 2013.

Feagin, R. A., Figlus, J., Zinnert, J. C., Sigren, J., Martínez, M. L., Silva, R., Smith, W. K., Cox, D., Young, D. R., and Carter, G.: Going with the flow or against the grain? The promise of vegetation for protecting beaches, dunes, and barrier islands from erosion, Frontiers in Ecology and the Environment, 13, 203–210, 2015.

Fiselier, J.: Milieueffectrapportage Zandmotor Delflandse kust, MER rapport door DHV in opdracht van provincie Zuid Holland, 2010.

Gemmell, A. R., Greig-Smith, P., and Gimingham, C. H.: A note on the behaviour of Ammophila arenaria (L.) Link, in relation to sand-dune formation, in: Transactions of the Botanical Society of Edinburgh, vol. 36, pp. 132–136, Taylor & Francis, 1953.

Hardisty, J.: Beach and nearshore sediment transport, Sediment transport and depositional processes. Blackwell, London, UK, pp. 216–255, 1994.

Hartigan, J. A. and Wong, M. A.: Algorithm AS 136: A k-means clustering algorithm, Journal of the Royal Statistical Society. Series C (Applied Statistics), 28, 100–108, 1979.

Hesp, P.: Foredunes and blowouts: initiation, geomorphology and dynamics, Geomorphology, 48, 245–268, 2002.

Hesp, P. A.: A review of biological and geomorphological processes involved in the initiation and development of incipient foredunes, Proceedings of the Royal Society of Edinburgh. Section B. Biological Sciences, 96, 181–201, 1989.

Hesp, P. A.: Ecological processes and plant adaptations on coastal dunes, Journal of arid environments, 21, 65–61, 1991.

Hesp, P. A. and Martínez, M. L.: Disturbance processes and dynamics in coastal dunes, Plant disturbance ecology: the process and the response, pp. 215–247, 2007.

Hilton, M. and Konlechner, T.: Incipient Foredunes Developed from Marine-dispersed Rhizome of Ammophilia arenaria, Journal of Coastal Research, p. 288, 2011.

Hoonhout, B. and De Vries, S.: Aeolian sediment supply at a mega nourishment, Coastal Engineering, 123, 11–20, 2017.

Houser, C., Hapke, C., and Hamilton, S.: Controls on coastal dune morphology, shoreline erosion and barrier island response to extreme storms, Geomorphology, 100, 223–240, 2008.

Houser, C., Wernette, P., Rentschlar, E., Jones, H., Hammond, B., and Trimble, S.: Post-storm beach and dune recovery: Implications for barrier island resilience, Geomorphology, 234, 54–63, 2015.

Huiskes, A. H. L.: Ammophila arenaria (L.) Link (Psamma arenaria (L.) Roem. et Schult.; Calamgrostis arenaria (L.) Roth), Journal of Ecology, 67, 363–382, 1979.

Huizer, S., Oude Essink, G. H., and Bierkens, M. F.: Fresh groundwater resources in a large sand replenishment, Hydrology and Earth System Sciences, 20, 3149–3166, 2016.

Jackson, N. L. and Nordstrom, K. F.: Aeolian sediment transport and landforms in managed coastal systems: a review, Aeolian research, 3, 181–196, 2011.

Jervey, M.: Quantitative geological modeling of siliciclastic rock sequences and their seismic expression, 1988.

Keijsers, J., De Groot, A., and Riksen, M.: Vegetation and sedimentation on coastal foredunes, Geomorphology, 228, 723–734, 2015a.

Keijsers, J. G. S., Giardino, A., Poortinga, A., Mulder, J. P. M., Riksen, M. J. P. M., and Santinelli, G.: Adaptation strategies to maintain dunes as flexible coastal flood defense in The Netherlands, Mitigation and Adaptation Strategies for Global Change, 20, 913–928, 2015b.

Keijsers, J. G. S., De Groot, A. V., and Riksen, M. J. P. M.: Modeling the biogeomorphic evolution of coastal dunes in response to climate change, Journal of Geophysical Research: Earth Surface, 2016.

Kelly, J. F.: Effects of human activities (raking, scraping, off-road vehicles) and natural resource protections on the spatial distribution of beach vegetation and related shoreline features in New Jersey, Journal of coastal conservation, 18, 383–398, 2014.

Konlechner, T. and Hilton, M.: The potential for marine dispersal of Ammophila arenaria (Marram Grass) rhizome in New Zealand, Journal of Coastal Research, pp. 434–437, 2009.

Konlechner, T. M., Hilton, M. J., and Orlovich, D. A.: Accommodation space limits plant invasion: Ammophila arenaria survival on New

Zealand beaches, Journal of coastal conservation, 17, 463–472, 2013.

Lithgow, D., Martínez, M., Gallego-Fernández, J., Hesp, P., Flores, P., Gachuz, S., Rodríguez-Revelo, N., Jiménez-Orocio, O., Mendoza-González, G., and Álvarez-Molina, L.: Linking restoration ecology with coastal dune restoration, Geomorphology, 199, 214–224, 2013.

Lu, D., Moran, E., and Batistella, M.: Linear mixture model applied to Amazonian vegetation classification, Remote sensing of environment, 87, 456–469, 2003.

Lucas, N. S., Shanmugam, S., and Barnsley, M.: Sub-pixel habitat mapping of a costal dune ecosystem, Applied Geography, 22, 253–270, 2002.

Luijendijk, A. P., Ranasinghe, R., de Schipper, M. A., Huisman, B. A., Swinkels, C. M., Walstra, D. J., and Stive, M. J.: The initial morphological response of the Sand Engine: A process-based modelling study, Coastal engineering, 119, 1–14, 2017.

[revised manuscript text omitted]

---

## Author Comment (AC2) · 11 Dec 2018

Dear Dr. Delgado-Fernandez,

Thank you for providing critical feedback to our manuscript. We attached a supplementary file with all the replies and the revisions applied to the text.

Sincerely, on behalf of the authors,

Corjan Nolet

Please also note the supplement to this comment:

https://www.earth-surf-dynam-discuss.net/esurf-2018-37/esurf-2018-37-AC2-supplement.pdf